# Comparison of transcriptional initiation by RNA polymerase II across eukaryotic species

Natalia Petrenko, Kevin Struhl*

Department of Biological Chemistry and Molecular Pharmacology, Harvard Medical School, Boston, United States

**ABSTRACT** The preinitiation complex (PIC) for transcriptional initiation by RNA polymerase (Pol) II is composed of general transcription factors that are highly conserved. However, analysis of ChIP-seq datasets reveals kinetic and compositional differences in the transcriptional initiation process among eukaryotic species. In yeast, Mediator associates strongly with activator proteins bound to enhancers, but it transiently associates with promoters in a form that lacks the kinase module. In contrast, in human, mouse, and fly cells, Mediator with its kinase module stably associates with promoters, but not with activator-binding sites. This suggests that yeast and metazoans differ in the nature of the dynamic bridge of Mediator between activators and Pol II and the composition of a stable inactive PIC-like entity. As in yeast, occupancies of TATA-binding protein (TBP) and TBP-associated factors (Tafs) at mammalian promoters are not strictly correlated. This suggests that within PICs, TFIID is not a monolithic entity, and multiple forms of TBP affect initiation at different classes of genes. TFIID in flies, but not yeast and mammals, interacts strongly at regions downstream of the initiation site, consistent with the importance of downstream promoter elements in that species. Lastly, Taf7 and the mammalian-specific Med26 subunit of Mediator also interact near the Pol II pause region downstream of the PIC, but only in subsets of genes and often not together. Species-specific differences in PIC structure and function are likely to affect how activators and repressors affect transcriptional activity.

**\*For correspondence:**
kevin@hms.harvard.edu

## Introduction

Transcription begins with the assembly of a preinitiation complex (PIC) at the promoter, a concept defined in vitro as a stable entity that contains RNA polymerase and initiates transcription upon addition of nucleotide triphosphates. For eukaryotic RNA polymerase (Pol) II, the PIC includes general transcription factors (GTFs), originally defined as being necessary and sufficient for 'basal' transcription from promoters in vitro (*Conaway and Conaway, 1993*; *Buratowski, 1994*; *Orphanides et al., 1996*; *Roeder, 1996*). GTFs are highly conserved among eukaryotic organisms, and they include the TATA-binding protein (TBP), TFIIA, TFIIB, TFIIE, TFIIF, TFIIH, and Pol II itself. However, variations in PIC composition can occur in metazoan tissues, including TBP-related factors (TRF1, -2, -3) that function primarily in reproductive organs (*D'Alessio et al., 2009*; *Vo Ngoc et al., 2017*).

Although the requirement for the GTFs for transcription in vitro can vary depending on reaction conditions, depletion experiments in yeast cells indicate that all GTFs are essential for Pol II transcription in vivo (*Petrenko et al., 2019*). Furthermore, the relative occupancies of GTFs are consistent across all yeast promoters and quantitatively linked to transcriptional activity (*Kuras et al., 2000*; *Pokholok et al., 2002*; *Rhee and Pugh, 2012*; *Petrenko et al., 2019*), indicating that a structurally similar PIC mediates a given level of transcription. Similar experiments in mammalian cells also suggest that a structurally similar PIC is responsible for most transcription (*Koch et al., 2011*; *Pugh and Venters, 2016*).

TBP-associated factors (TAFs), which together with TBP constitute the TFIID complex, are not required for transcription in vitro, but they are often components of the PIC in vivo. In yeast cells, TAF occupancies are strongly correlated with each other, but they do not strictly correlate with TBP occupancy (*Kuras et al., 2000*; *Li et al., 2000*; *Rhee and Pugh, 2012*). Thus, transcription in yeast cells can be mediated by PICs with TAF-containing (TFIID) or TAF-lacking forms of transcriptionally active TBP. The relative usage of these two PIC forms depends on the promoter and the quality of the TATA sequence (*Struhl, 1986*; *Iyer and Struhl, 1995*; *Moqtaderi et al., 1996*; *Basehoar et al., 2004*; *Huisinga and Pugh, 2004*; *Petrenko et al., 2019*) as well as the activator protein that stimulates PIC formation (*Li et al., 2002*; *Mencia et al., 2002*). The TAF-lacking form of TBP may be associated with the SAGA complex via the Spt3 subunit (*Eisenmann et al., 1992*; *Bhaumik and Green, 2001*; *Larschan and Winston, 2001*; *Basehoar et al., 2004*; *Papai et al., 2020*; *Wang et al., 2020*), but there is no direct evidence that SAGA associates with the promoter in vivo.

Some metazoan tissues possess tissue-specific TAFs and tissue-specific TFIID subunit composition (*D'Alessio et al., 2009*; *Hart et al., 2009*; *Maston et al., 2012*). There is no evidence for the existence of free TBP in metazoan cells, and it is unclear whether there are TAF-lacking, or perhaps SAGA-containing forms of transcriptionally active TBP. However, fluorescent studies suggest that activity of *Drosophila* histone gene promoters might rely only on TBP and TFIIA in the absence of TFIIB and TAFs (*Guglielmi et al., 2013*).

Mediator is a large complex (21–30 proteins depending on the species) that interacts with Pol II, and it can stimulate PIC assembly, phosphorylation of the Pol II C-terminal domain (CTD) by TFIIH, and basal transcription in vitro (*Thompson et al., 1993*; *Kim et al., 1994*; *Guidi et al., 2004*; *Takagi and Kornberg, 2006*; *Esnault et al., 2008*). Mediator is organized structurally into head, middle, tail, and kinase modules (*Guglielmi et al., 2004*; *Lariviere et al., 2012*; *Allen and Taatjes, 2015*; *Plaschka et al., 2015*; *Robinson et al., 2015*). The evolutionarily divergent tail module interacts with activator proteins bound at enhancers, whereas the highly conserved head and middle modules interact with Pol II. In yeast, the complete Mediator complex is essential for transcription, but sub-modules can support transcription at reduced levels (*Petrenko et al., 2017*). The kinase module sterically blocks the interaction of Mediator with Pol II (*Elmlund et al., 2006*; *Knuesel et al., 2009*), but it has a very modest effect on transcription.

In wild-type yeast cells, Mediator is readily detected at enhancers but is not detected at promoters (*Fan et al., 2006*), even though it is a required component of the PIC (*Jeronimo and Robert, 2014*; *Wong et al., 2014*). In contrast, Mediator in mammalian cells associates with both promoters and enhancers (*Kagey et al., 2010*). Nevertheless, in yeast, Mediator associates strongly with promoters upon depletion or inhibition of TFIIH kinase (Kin28 subunit), and the level of Mediator association strongly correlates with GTF occupancy and Pol II transcription (*Jeronimo and Robert, 2014*; *Wong et al., 2014*). Although the complete Mediator complex is recruited by activator proteins bound to enhancers upstream of the promoter, the form of Mediator associated with the promoter lacks the kinase module. This observation is in accord with structural and biochemical studies showing that the kinase module inhibits Mediator contacts with Pol II. Thus, a single Mediator complex acts as a dynamic bridge between enhancers and promoters, and it undergoes a compositional change in which the kinase module dissociates to permit association with Pol II and the PIC (*Jeronimo et al., 2016*; *Petrenko et al., 2016*).

These observations also indicate that, in wild-type yeast cells, Mediator association with the promoter is transient, and hence the PIC is a very short-lived entity (estimate around 1/8 s) (*Wong et al., 2014*). Upon PIC formation, TFIIH kinase rapidly phosphorylates the Pol II CTD at serine 5 with concomitant dissociation of Mediator and Pol II escape from the promoter (*Jeronimo and Robert, 2014*; *Wong et al., 2014*). As the PIC is transient, measurements of GTF occupancy in wild-type cells largely reflect a complex in which Pol II has escaped but which is suitable for re-association of Mediator and a new Pol II molecule (*Wong et al., 2014*). It has been suggested that this post-escape complex is sufficiently stable to permit multiple rounds of reinitiation (also termed bursting) without having to form a completely new PIC from scratch (*Wong et al., 2014*).

In metazoans, while some aspects of transcription have received intense scrutiny (e.g., Pol II pausing downstream of the promoter, divergent transcription, enhancer RNAs), issues related to basic initiation mechanisms in vivo are largely unaddressed. Here, we analyze publicly available (as of September 2021) genome-wide data for PIC components in metazoans, focusing on cell lines with ChIP-seq

datasets for multiple factors to allow direct comparison. We show yeast and metazoans differ considerably with respect to Mediator function, PIC stability, and transcriptional initiation. We show that TFIID is not a monolithic entity at all promoters and extend previous studies indicating that TAFs, and particularly Taf7, behave differently in yeast and metazoans, and we investigate the complicated relationship between Taf7 and Med26.

## Results

### Promoter definition

The original, and still the best, definition of a promoter is the genetic element necessary for expression of a structural gene (or operon) that is distinct from elements that regulate the expression level of the gene (*Jacob et al., 1964*; *Scaife and Beckwith, 1966*). In molecular terms, promoters are recognized by basic transcription machineries that initiate RNA synthesis from nearby sites. However, over the past few decades, this fundamental concept of promoters has become muddled.

A common and confusing definition, particularly for metazoan promoters, includes proximal DNA sequences recognized by specific transcription factors that vary among genes linked to what is termed the 'core promoter'. Another source of confusion involves the term 'enhancer', originally defined as a genetic element that interacts with specific activator proteins that stimulate transcription from a separable promoter located far away (*Banerji et al., 1981*; *Moreau et al., 1981*; *Fromm and Berg, 1982*). However, enhancers are bound by the same sequence-specific activator proteins that bind to promoter-proximal regions, and they often express 'enhancer RNAs', leading to the confusing idea that enhancers and promoters are very similar (*Core et al., 2014*).

In this paper, which relies exclusively on ChIP-seq datasets, we define a promoter by occupancy of GTFs (typically TBP) and Pol II (typically observed at the pause site, not the promoter itself). Many promoters are located immediately upstream of mRNA coding regions, and these mRNA promoters are the primary focus of the analyses. However, other promoters are in the vicinity of what are termed enhancer regions. As we will show, these two classes of promoters behave indistinguishably from the perspective of PIC function. Promoters defined by the classic concept are distinct from activator-binding sites, irrespective of their distance from a given promoter.

### Mammalian Mediator associates stably with promoters, not activator-binding sites at proximal or distal locations

In wild-type *Saccharomyces cerevisiae* cells, Mediator associates strongly with active enhancers but not with promoters (*Fan et al., 2006*; *Fan and Struhl, 2009*; *Figure 1A*). However, Mediator association with promoters does occur in strains depleted for the TFIIH kinase activity (*Jeronimo and Robert, 2014*; *Wong et al., 2014*; *Jeronimo et al., 2016*; *Petrenko et al., 2016*; *Figure 1A*). Furthermore, unlike the case at enhancers, the form of Mediator at promoters lacks the kinase module (*Jeronimo et al., 2016*; *Petrenko et al., 2016*; *Figure 1A*).

By contrast, in wild-type human and mouse cells, Mediator (defined by the middle module subunit Med1) binds to promoters of protein-coding genes (*Kagey et al., 2010*; *Figure 1B and C*). Med1 peaks overlap closely with those of TBP, TFIIB, and TFIIF, but all these sites are about 80 bp downstream of peaks of transcription factors SP1 and NF-Y, which are markers of promoter-proximal enhancers just upstream of promoters (*Figure 1B* and *Figure 1—figure supplement 1A*). Med1 and GTF occupancy sites are also about 50 bp upstream of Pol II, which is located at the post-initiation pause position as expected (*Figure 1B* and *Figure 1—figure supplement 1A*). The distinct peak locations of Pol II, GTFs, and transcription factors in the same samples indicate that Mediator is associated with the promoter and not with the promoter-proximal elements bound by activator proteins that stimulate basal transcription.

Distal regulatory regions, typically termed enhancers, can support a low level of Pol II transcription that generates non-coding 'enhancer RNAs' (*Kim et al., 2010*; *Core et al., 2014*; *Jin et al., 2017*). Such enhancer regions are often defined by chromatin that is accessible (DNase-sensitive) and flanked by acetylated histones (*Heintzman et al., 2007*; *Ernst et al., 2011*), and mammalian cells typically have ~150,000 such enhancers (*Shen et al., 2012*; *ENCODE, 2020*). These chromatin features at enhancers are mediated by sequence-specific activator proteins that cooperatively recruit nucleosome remodeling and histone acetylase complexes.

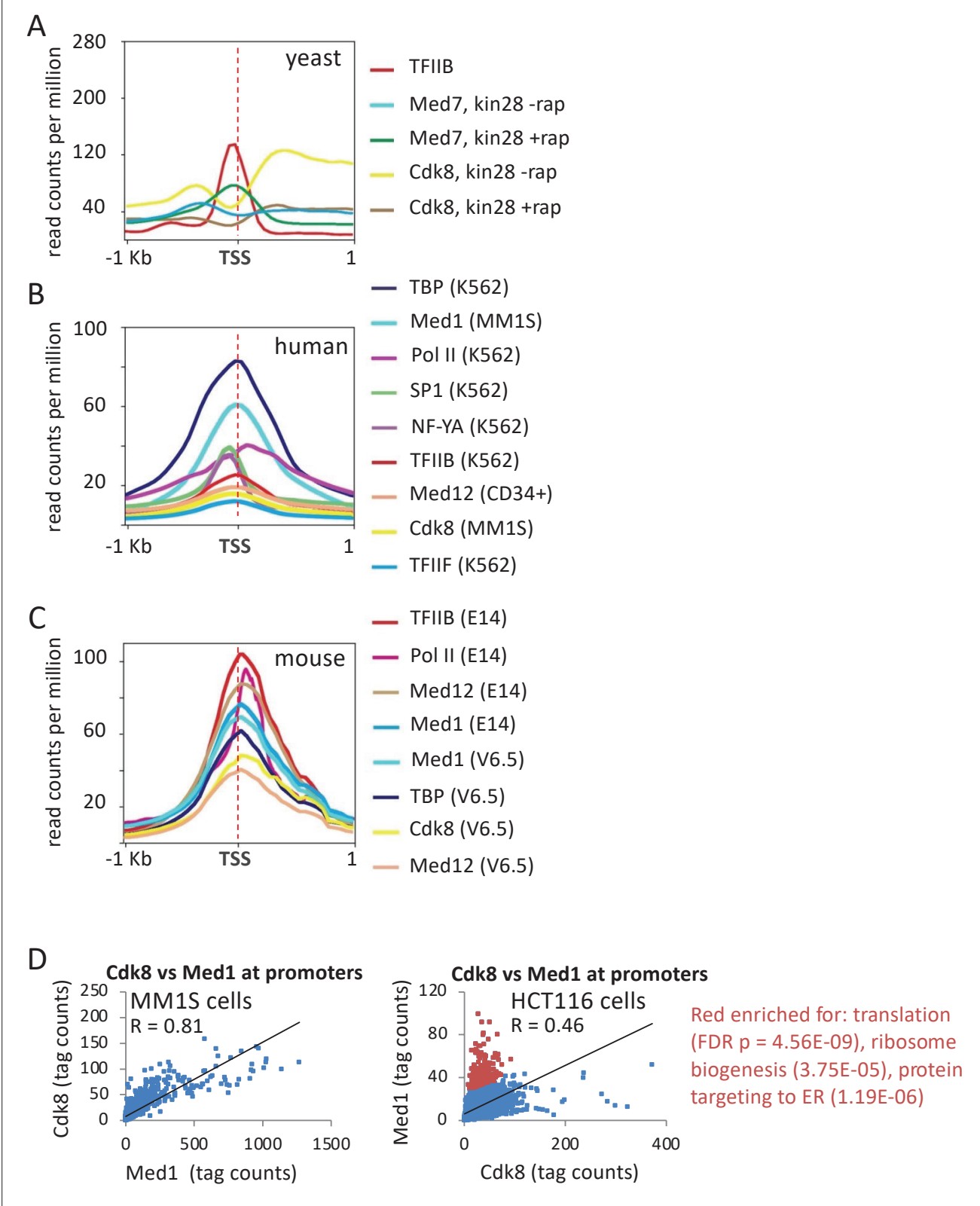

**Figure 1.** Association of general transcription factors (GTFs) at mRNA promoters and distal regulatory regions. (**A**) Mean occupancies (metagene analyses) of the indicated proteins at the 600 most active yeast mRNA promoters with respect to the transcription start site (TSS) in cells depleted (+rap) or not depleted (-rap) for the Kin28 subunit of TFIIH. (**B**) Mean occupancies of the indicated proteins at the 10,000 most active mRNA promoters (by TATA-binding protein [TBP] occupancy) in the indicated human cell lines. (**C**) Mean occupancies of the indicated proteins at the 10,000 most active

*Figure 1 continued on next page*

*Figure 1 continued*

mRNA promoters in the indicated mouse cell lines. (**D**) Correlation of Med1 and Cdk8 occupancies at individual mRNA promoters (dots) in MM1S and HCT116 cells; red dots indicate promoters with relatively high Med1 levels with respect to Cdk8 levels.

The online version of this article includes the following figure supplement(s) for figure 1:

**Figure supplement 1.** Association of general transcription factors (GTFs) at promoters in human cells.

The level of Mediator association (median values of Med1 peak heights) at a small subset of such enhancer regions (top 10,000 Mediator peaks out of ~150,000 enhancers) is roughly comparable to the level of Mediator association at mRNA promoters (top 10,000 or all Mediator peaks at ~20,000 mRNA promoters) (*Figure 2A*). In striking contrast, the median level of Mediator association at all enhancers is much lower (*Figure 2A*). Moreover, most of the apparent Mediator occupancy at all enhancers is due to contributions from the top 10,000 enhancers. Thus, the vast majority of enhancers have low or non-detectable levels of Mediator.

Distinguishing whether Mediator binding within the so-called enhancer regions coincides with activator-binding sites or the PIC is complicated for several reasons. First, it is difficult to align enhancers, unlike promoters that are aligned by their transcription start. Second, transcription from enhancer regions is largely bidirectional, making it difficult to distinguish upstream from downstream relationships between associated proteins. Third, the PICs responsible for the divergent enhancer RNAs are often too close together to be resolved separately. In these cases, the GTF peaks appear to map to the center of the enhancer, which coincides with the activator peaks, thereby making it impossible to address whether Mediator is recruited by the activator or the PIC/promoter.

For these reasons, we measured the distance between the peak summits of Mediator (Med1 subunit), the GTF TFIIH (Cdk7 subunit), Pol II, and sequence-specific activators (E2F, DP1) at enhancers in MM1S cells (*Figure 2B*). As a control, the median distance for an individual factor in biological replicates is ~45 bp, reflecting experimental variation in peak summits. The median distance between Mediator and Cdk7 is similar, confirming that Mediator coincides with the PIC. The median distance between Cdk7 and Pol II is ~90 bp, indicating that Pol II is at the paused position even at the so-called enhancers. Importantly, the median distance between Mediator and the E2F and DP1 activators is ~90 bp, indicating that Mediator is not localized at the same position as the activators. As expected, similar results are observed at mRNA promoters (*Figure 2B*). Thus, even at the so-called enhancers, Mediator is stably associated with promoters, not activator-binding sites.

## The kinase module of mammalian Mediator associates with promoters

Unlike the situation at yeast promoters, the kinase module (represented by the Cdk8 and Med12 subunits) associates with human and mouse promoters (*Figure 1B and C*). Association of the kinase module at the promoter is strongly correlated with Med1 for most genes, although the kinase module is depleted at certain classes of genes in some cell lines (translation, ribosome biogenesis and protein targeting to the endoplasmic reticulum in HCT116 cells, chromatin organization in MOLM-14 cells, and histone genes in mouse V6.5 cells; *Figure 1D* and *Figure 1—figure supplement 1B*). Thus, mammalian Mediator stably associates with the promoter in a manner that can include the kinase module. The presence of the kinase module at mammalian promoters is surprising in light of numerous lines of evidence indicating that the kinase module and Pol II interact with the core Mediator in a mutually exclusive manner (see Discussion).

## Mediator and PIC levels are strongly, but not strictly, correlated at promoters

To address whether Mediator and PIC components co-associate with the promoter, we compared the levels of Med1, Pol II, and TBP at the 10,000 most active mRNA promoters (those with highest TBP levels). A strong Pearson correlation (range 0.65–0.74) suggests a relatively constant ratio of Mediator and GTFs at the promoter for many genes (*Figure 3A* and *Figure 3—figure supplement 1*), as is the case in yeast cells. However, Mediator levels are higher than those of PIC components at cell line-specific categories of genes (*Figure 3A*), excluding the possibility that the relatively high Mediator levels at these genes are due to experimental variance. Promoters with relatively increased Med1 and Cdk8 are enriched for chromatin organization and immunity-related genes in the MM1S blood

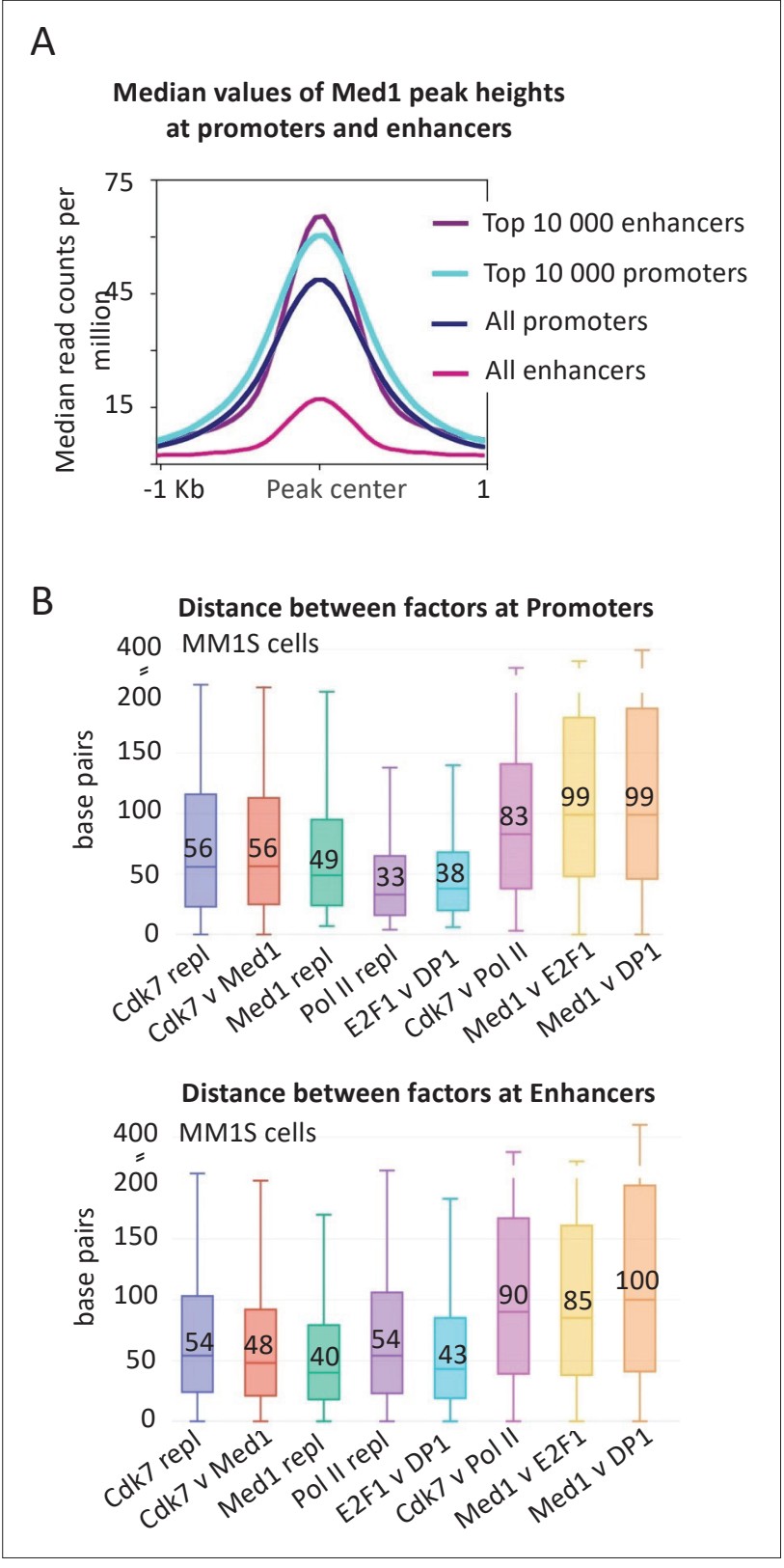

**Figure 2.** Mediator association and distance between factors at promoters and enhancers. (**A**) Medan occupancy values at and surrounding the preinitiation complex (PIC) peak center defined by general transcription factor (GTF occupancy) of the top 10,000 promoters and enhancers as well as all promoters and enhancers in MM1S cells. All enhancers include the ~60,000 genomic regions where any Mediator signal is detected, including the top 10,000.

*Figure 2 continued on next page*

*Figure 2 continued*

It does not include the ~100,000 enhancers where no Mediator signal is detected. (**B**) Absolute values of the distances in base pairs betwen the indicated factors (or replicates) at top 10,000 promoters and enhancers (*Loven et al., 2013*) in MM1S cells. The boxplot indicates the median value (also written as a number), as well as the 25th and 75th percentiles, represented as the edges of the box.

cancer cell line, for differentiation and apoptosis genes in human embryonic stem cells (hESCs), and stress responses in HCT116 cells (*Figure 3A*). Thus, unlike the case in yeast cells depleted of Kin28, mammalian Mediator association with promoters is not strictly correlated with association of other PIC components.

We also examined the relationship between Mediator, Cdk7, and Pol II at distal enhancers in MM1S cells (*Figure 3B*). The correlation of Mediator occupancy with either Cdk7 and Pol II occupancy is strong ($R$ = 0.6) and only slightly less than the correlation between the PIC components Cdk7 and Pol II ($R$ = 0.7) at distal enhancers and the correlations at mRNA promoters (*Figure 3A*). In accord with the median Mediator occupancy at enhancers (*Figure 2A*), but in marked contrast to the situation in yeast, there are few enhancers in which Mediator is associated but GTFs are not (*Figure 3B*).

## Fly Mediator associates stably with promoters and may vary in composition

*Drosophila melanogaster* Med1 (middle module) and Med30 (part of a tail segment contiguous with the head) (*El Khattabi et al., 2019*) co-localize with TBP at the promoter (*Figure 4A*). As in mouse and human cells, the Mediator peak maps ~60 bp upstream of the Pol II peak and ~60 bp downstream of the peak for the GAGA transcription factor, indicating association with the PIC, not promoter-proximal elements for sequence-specific DNA-binding proteins. Unexpectedly, some genes show high levels of Med1 but nearly absent Med30, or vice versa (*Figure 4B* and *Figure 4—figure supplement 1*). This discordance between Med1 and Med30 binding is not due to random errors, because each set of genes shows enrichment in gene ontology (GO) categories (*Figure 4B*). Thus, while Mediator appears to behave as a complete complex at most fly promoters, some promoters appear to be bound by distinct Mediator subcomplexes and/or different conformations of Mediator with altered crosslinking properties. These distinct forms of Mediator might be involved in tissue and gene specificity of essential Mediator subunits in other organisms. For example, mouse Med9 (middle module) is essential in T cells but not in B cells or ESCs, whereas Med26 (middle module) is essential in T cells and ESCs but not in B cells (*El Khattabi et al., 2019*).

## Pol II occupancy at 5' ends of genes, but not in the coding region, is strongly correlated with PIC occupancy

In yeast, Pol II occupancy in the coding regions directly corresponds to the occupancy of PIC components at the promoter, as Pol II rapidly escapes the promoter into active elongation. By contrast, in mammalian and fly cells, Pol II occupancy peaks at the post-initiation pause site ~50 bp downstream of the transcription start site (TSS), and it is considerably lower throughout the coding region (*Adelman and Lis, 2012*; *Core and Adelman, 2019*). Paused Pol II is released into elongation via phosphorylation of NELF and Spt5 by the Cdk9 subunit of the pTEFb complex. In addition, paused Pol II can be removed through early termination, in a mechanism involving the Integrator complex (*Erickson et al., 2018*; *Elrod et al., 2019*; *Huang et al., 2020*). Paused Pol II inhibits new Pol II initiation because it sterically blocks Pol II association at the promoter and hence assembly of a functional PIC (*Gressel et al., 2017*; *Shao and Zeitlinger, 2017*).

Resolving Pol II occupancy at the promoter versus the pause site requires high-resolution (e.g., ChIP-exo) or nucleotide-level (e.g., Pro-seq) data that is not available in cell lines for which there is data for other initiation factors. Thus, we used total Pol II occupancy level at peaks around the promoters to determine its relationship to TBP levels. If levels of paused Pol II relative to the initiation rate vary across the genome, then there should be appreciable deviations of the ratio of Pol II occupancy relative to other PIC components. However, Pol II levels correlate very strongly with TBP levels (Pearson = 0.8 in K562 and hESC) (*Figure 5A*), albeit slightly below the correlation (0.9–0.96) of biological replicates (*Figure 5—figure supplement 1*). This indicates that Pol II occupancy near promoters (which is primarily at the pause site) is strongly linked to PIC levels. In addition, these

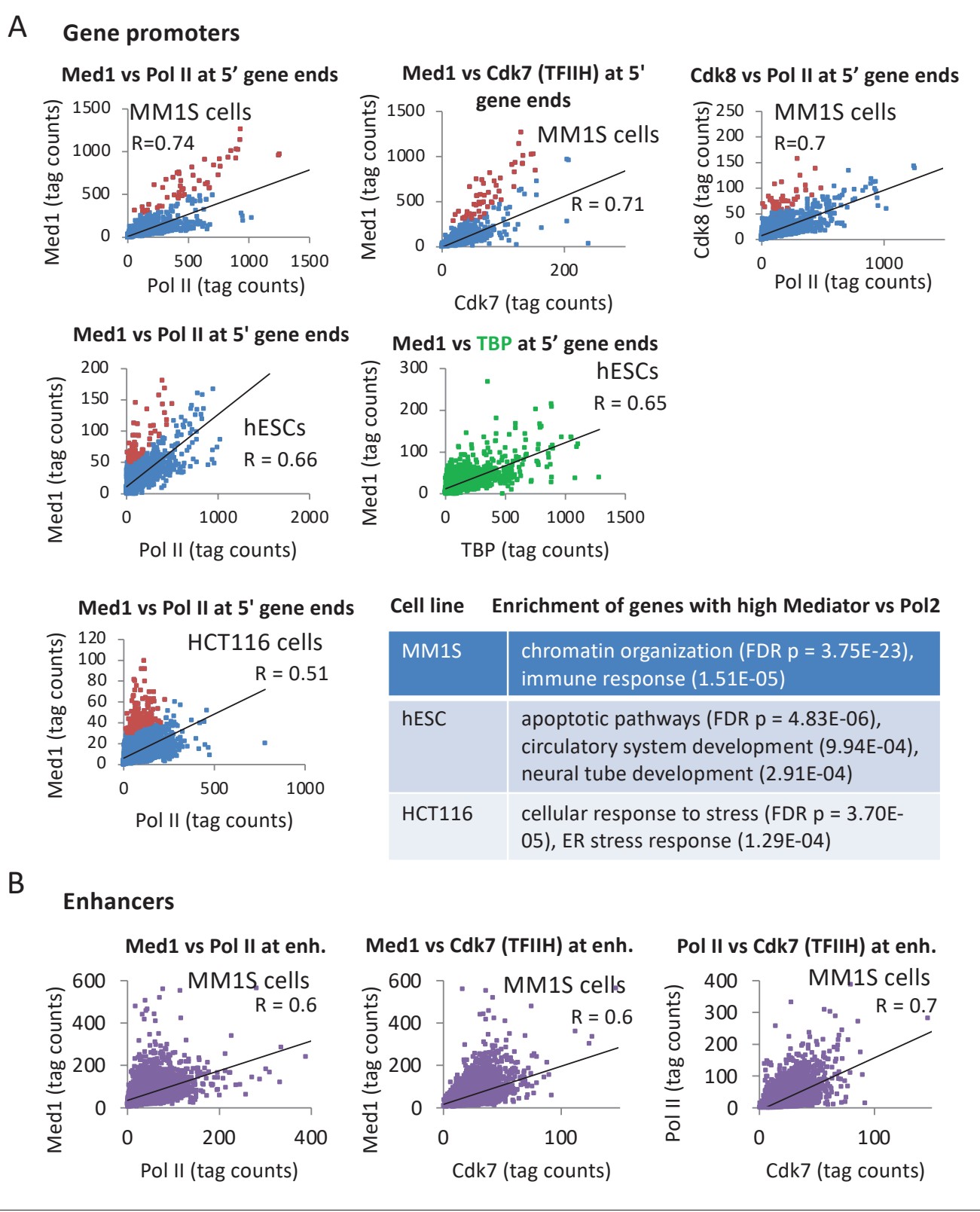

**Figure 3.** Relationship of Mediator, TATA-binding protein (TBP), and Pol II levels at promoters. (**A**) Pairwise correlations of Med1, TBP, and Pol II occupancies at the 10,000 most active (by TBP occupancy) mRNA promoters (dots) in the indicated human cell lines; red dots indicate promoters with relatively high Mediator levels with respect to Pol II levels. Enriched gene categories for promoters with relatively high Mediator levels with respect to Pol II levels in the indicated cell lines are given in the *Supplementary file 2*. (**B**) Pairwise correlations of Med1, Cdk7 (TFIIH), and Pol II occupancies

*Figure 3 continued on next page*

*Figure 3 continued*

at ~8000 enhancers (*Loven et al., 2013*) in MM1S cells.

The online version of this article includes the following figure supplement(s) for figure 3:

**Figure supplement 1.** Pairwise correlations of Cdk7 (TFIIH) and Pol II occupancies at the 10,000 most active mRNA promoters (dots) in MM1S cells.

observations argue that premature termination at the pause site either happens in concert with initiation or is not a major mechanism regulating Pol II activity at most genes.

As expected from the fact that phosphorylation of the Pol II CTD at Serine 5 is mediated by the Cdk7 subunit of TFIIH, there is a very strong correlation between the levels of total Pol II and CTD-S5-phosphorylated Pol II (Pearson = 0.87; *Figure 5B*). However, the correlation between total Pol II and S2-phosphorylated Pol II is lower (Pearson = 0.67; *Figure 5C*), possibly reflecting the competition between initiation mechanisms and the activities of pausing factors and Integrator. In accord with the observation that very highly transcribed mouse genes often have lower relative levels of pausing (*Min et al., 2011*), there is a tendency for decreased pausing index (the ratio of Pol II in the promoter-proximal peak relative to the coding region) to be linked to increased TBP occupancy (*Figure 5D*).

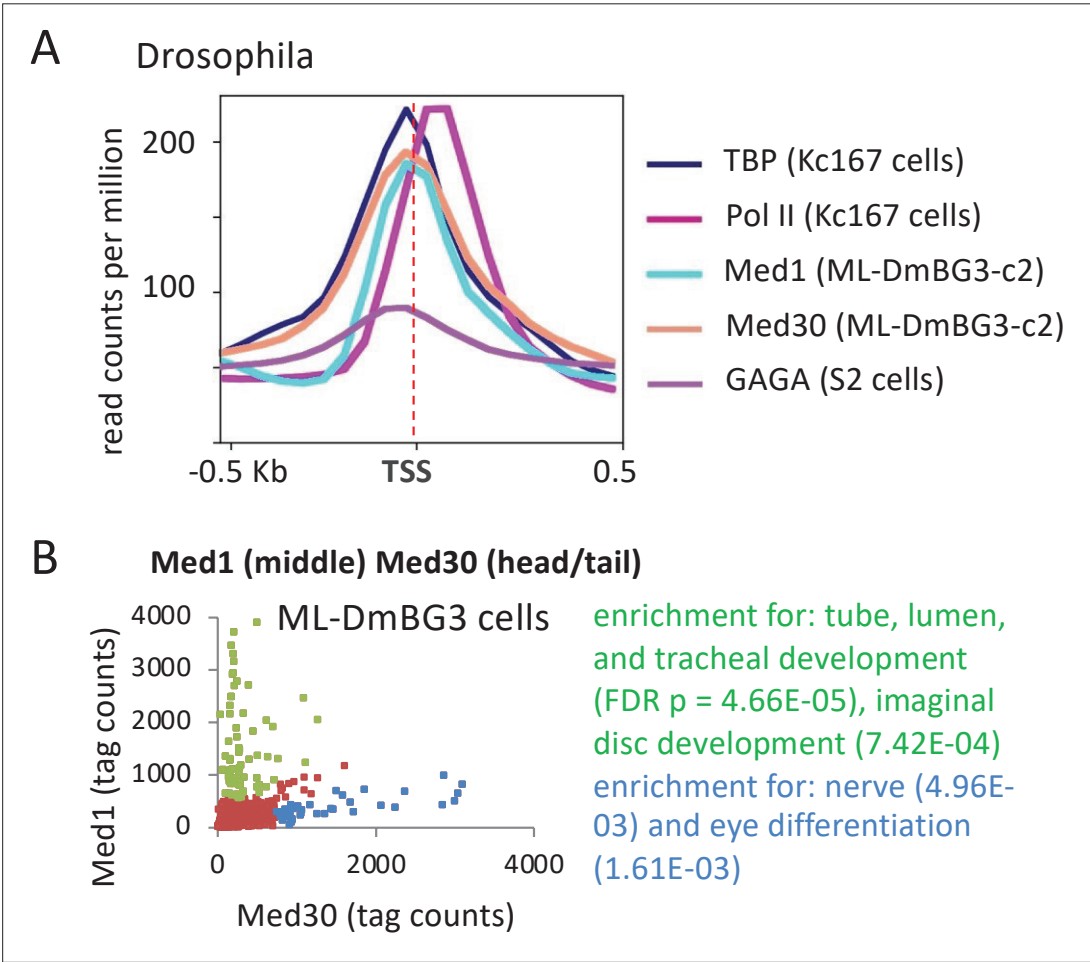

**Figure 4.** Association of Mediator, TATA-binding protein (TBP), Pol II, and GAGA transcription factor in *Drosophila* cell lines. (**A**) Mean occupancies of the TBP, Pol II, Mediator (Med1 and Med30 subunits), and GAGA at the 1000 most active (by TBP occupancy) promoters in the indicated *Drosophila* cell lines with respect to the transcription start site (TSS). (**B**) Relative occupancy levels of Med30 and Med1 at individual promoters. Genes with relatively high Med30:Med1 ratios (blue dots) and relatively low Med30:Med1 ratios (green dots) indicated along with enriched gene categories.

The online version of this article includes the following figure supplement(s) for figure 4:

**Figure supplement 1.** Screenshots of Med1 and Med30 occupancy levels at several fly genes in ML-DmBG3 cells.

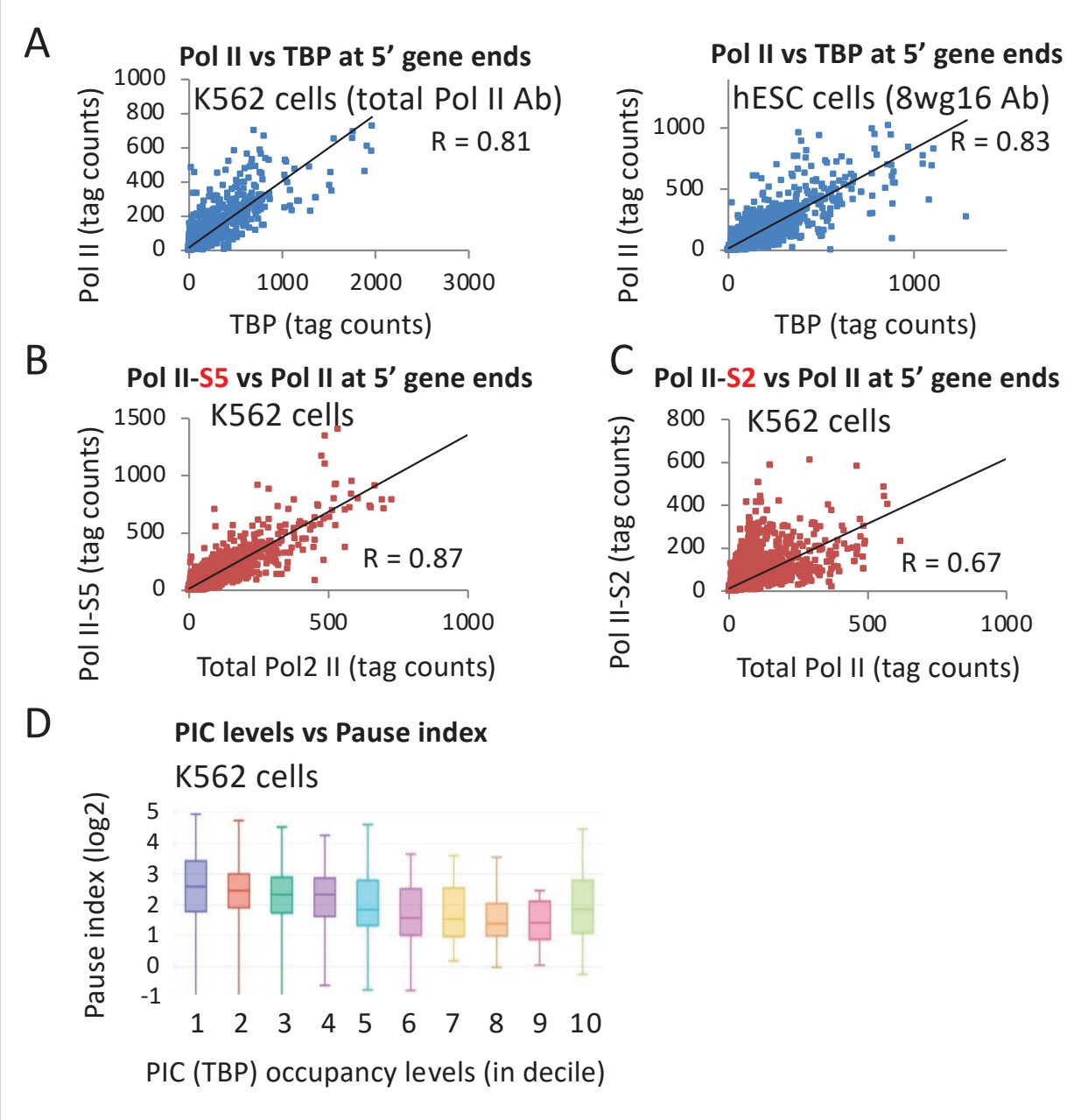

**Figure 5.** Relationship of TATA-binding protein (TBP) occupancy with various forms of Pol II. (**A**) Relative occupancy levels of TBP and total Pol II in the vicinity of individual promoters (10,000 most active by TBP occupancy) in the indicated human cell lines. (**B**) Relative occupancy levels of total Pol II and the form of Pol II phosphorylated at serine 5 in the C-terminal domain in the vicinity of individual promoters in K562 cells. (**C**) Relative occupancy levels of total Pol II and the form of Pol II phosphorylated at serine 2 in the C-terminal domain in the vicinity of individual promoters in K562 cells. (**D**) Relationship between TBP occupancy levels subdivided into deciles, in which high values indicate high occupancy with the Pol II pausing index (ratio of Pol II in the promoter-proximal peak relative to the coding region).

The online version of this article includes the following figure supplement(s) for figure 5:

**Figure supplement 1.** Correlations of Pol II occupancies in biological replicates and datasets from different laboratories.

## TFIID dependency varies across metazoan promoters

In yeast cells, TBP and the TAF subunits of TFIID have indistinguishable binding profiles at promoters. However, levels of TAF occupancies do not strictly correlate with TBP or GTF occupancies, leading to the concept of TAF-containing (TFIID) and TAF-lacking forms of the PIC transcriptionally active TBP (*Kuras et al., 2000*; *Li et al., 2000*; *Rhee and Pugh, 2012*). Promoters favored by TFIID-containing

PICs tend to have lower quality TATA motifs, consistent with the idea that TAF interactions with promoter DNA can reduce the requirement for a strong TBP:TATA interaction. In TAF-depleted cells, transcriptionally active PICs lacking TAFs have been directly observed (*Petrenko et al., 2019*).

In flies, unlike in yeast and human, TAF1 and TAF2 show well-defined binding peaks ~60 bp downstream of TBP (*Baumann and Gilmour, 2017*; *Shao and Zeitlinger, 2017*; *Figure 6A*). This observation is consistent with the location of downstream promoter elements (e.g., DPE and MTE) that are important for transcription and directly contacted by TAF1 and TAF2 (*Louder et al., 2016*; *Baumann and Gilmour, 2017*; *Vo Ngoc et al., 2019*). In agreement with a pared-down PIC lacking TAFs and several GTFs at the histone gene clusters (*Guglielmi et al., 2013*), TAF1 and TAF2 are completely missing despite considerable association of TBP and Pol II (*Figure 6—figure supplement 1A*).

Yeast genes involved in translation and housekeeping functions tend to show a greater reliance on TFIID, whereas highly regulated genes tend to be TAF-independent and dependent on the SAGA complex (*Moqtaderi et al., 1996*; *Kuras et al., 2000*; *Li et al., 2000*; *Huisinga and Pugh, 2004*; *de Jonge et al., 2017*; *Petrenko et al., 2019*). Similarly, the Taf2:TBP ratio in fly cells varies significantly more than observed for the TBP:TBP ratio of replicates (*Figure 6B* and *Figure 6—figure supplement 1B*). Moreover, genes with high or low Taf2:TBP ratios are enriched for certain gene classes (*Figure 6B* and *Figure 6—figure supplement 1B*), indicating that variations in this ratio are not due to chance. Genes with a high Taf2:TBP ratio tend to be involved in translation, protein targeting to membrane, and nonsense mediated decay, whereas those with a low Taf2:TBP ratio are involved in metabolism and organ morphogenesis (*Figure 6B*). The observation that high and low TAF/TBP occupancy ratios are respectively enriched for housekeeping and developmental genes in human cells resembles what occurs in yeast.

In human cells, the binding profile of TAF1 is indistinguishable from that of TBP (*Figure 6A*). Taf1 occupancy correlates well with TBP occupancy (Pearson = 0.64 in K562 and 0.78 in hESC *Figure 6—figure supplement 2A*), but some promoters have low Taf1 occupancy relative to TBP occupancy (*Figure 6C*). Promoters with relatively low Taf1 occupancy constitute ~20 % of those with detectable TBP peaks, and they are enriched for nucleosome organization (including histones), chromatin assembly, and certain stress responses in two different cell lines, while those with the highest Taf1:TBP occupancy ratio are enriched for translation, ribosome biogenesis, and cell cycle-related processes (*Figure 6C* and *Figure 6—figure supplement 2B*). In contrast and as expected, TFIIB:TBP occupancy ratios show less variation, and the few promoters with high or low ratios are not enriched for any functional categories and are likely due to experimental variation (*Figure 6C* and *Figure 6—figure supplement 2A*).

## Discordant behavior of Taf1 and Taf7 at distinct sets of human genes

In *S. cerevisiae*, all TAFs tested maintain a constant ratio to one another across various genes (*Kuras et al., 2000*; *Li et al., 2000*; *Venters et al., 2011*), suggesting that TFIID functions as a monolithic complex. Although TAF7 has not been investigated in *S. cerevisiae*, TBP and Taf7 occupancy in *Schizosaccharomyces pombe* is strongly correlated (Pearson = 0.7) at promoters of genes encoding proteins (*Figure 7A*) and non-coding RNAs (*Figure 7B*). Interestingly, *S. pombe* genes with relatively high Taf7 levels are enriched for translation and ribosome biogenesis (*Figure 7A*), resembling the relatively high TAF levels and TFIID dependency at such genes in *S. cerevisiae*.

In contrast, human TAF7 is not only present at the PIC, but it also appears to associate downstream of human MHC class I promoters in a manner independent of other TFIID components (*Gegonne et al., 2008*). Taf7 has been suggested to negatively regulate Taf1 (*Gegonne et al., 2001*), to suppress TFIIH and p-TEFb kinases (*Gegonne et al., 2008*), to dissociate from the PIC, and to act as a checkpoint for preventing premature initiation and elongation (*Gegonne et al., 2006*; *Gegonne et al., 2013*). It is unknown whether these observations are limited to the few genes tested or are more general.

Genome-scale analysis of Taf1 and Taf7 occupancy in K562 and hESC cells reveals a complex picture. Like TAF1, Taf7 occupancy correlates reasonably well with TBP (Pearson = 0.57 in K562 and 0.72 in hESC; *Figure 7—figure supplement 1A*). However, some promoters display stronger than expected Taf7 occupancy relative to TBP occupancy (*Figure 7C*; compare with Taf1 replicates in *Figure 7—figure supplement 1B*). Interestingly, these high Taf7 promoters are enriched for genes involved in nucleosome organization, chromatin assembly, and DNA replication (*Figure 7C*), the same

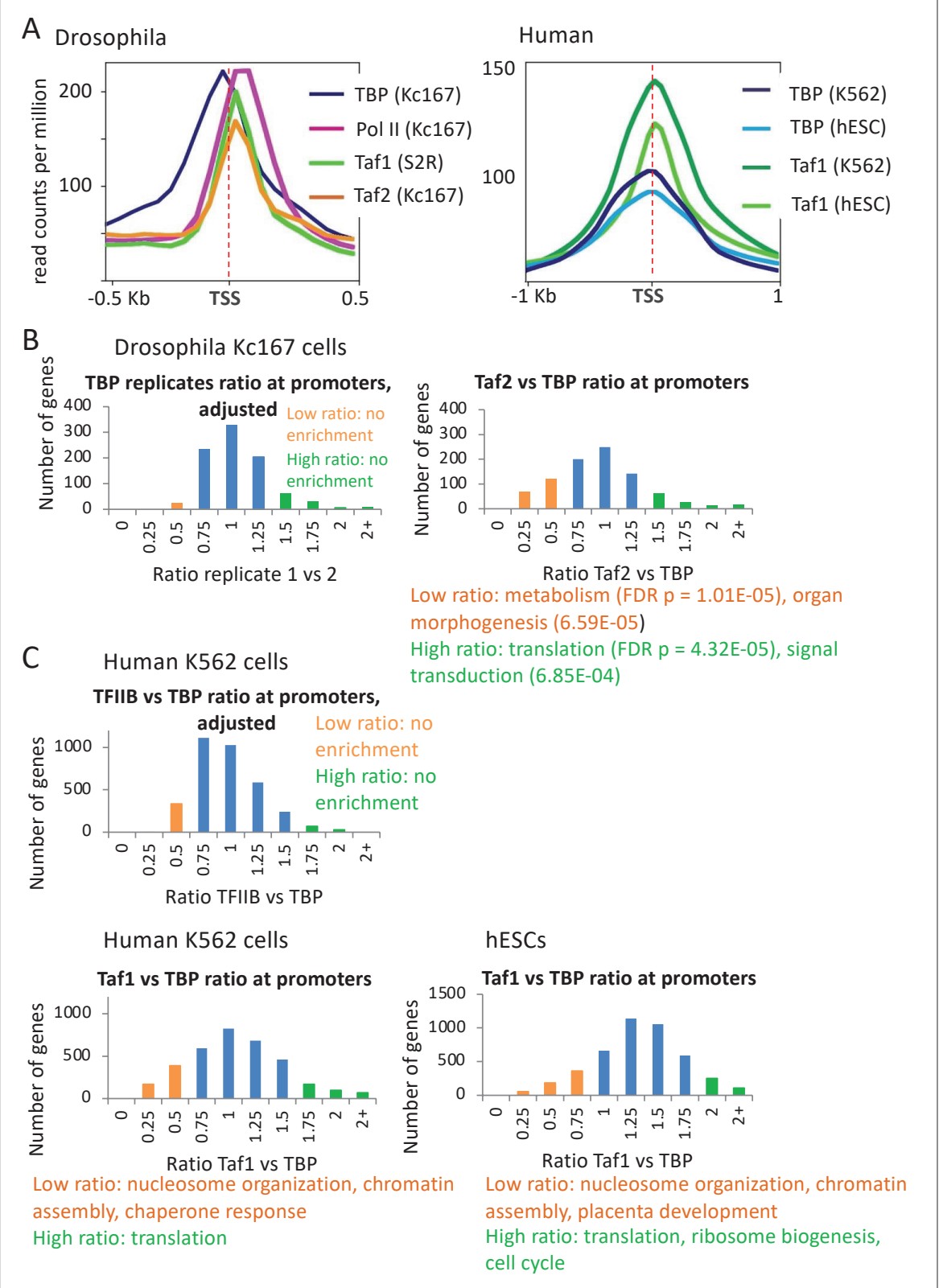

**Figure 6.** Relative occupancy of TBP-associated factors (Tafs) and TATA-binding protein (TBP) at promoters. (**A**) Relative occupancy levels of TBP, Pol II, and the indicated Tafs at the 1000 most active *Drosophila* and 10,000 most active human mRNA promoters with respect to the transcription start site (TSS). (**B**) Number of *Drosophila* mRNA promoters having the indicated TBP:TBP replicate or Taf2:TBP ratios (among 900 most active promoters); median ratio set to 1.0. Promoters with low (orange) or high (green) Taf2:TBP ratios indicated along with enriched gene categories. (**C**) Number of

*Figure 6 continued on next page*

*Figure 6 continued*

human promoters having the indicated TFIIB:TBP or Taf1:TBP ratios (among 3500 most active promoters) in the indicated cell lines; median ratio set to 1.0. Promoters with low (orange) or high (green) Taf1:TBP ratios indicated along with enriched gene categories.

The online version of this article includes the following figure supplement(s) for figure 6:

**Figure supplement 1.** Relative occupancy of TBP-associated factors (Tafs) and TATA-binding protein (TBP) at promoters in *Drosophila*.

**Figure supplement 2.** Relative occupancy of TBP-associated factors (Tafs) and TATA-binding protein (TBP) at promoters in human cell lines.

categories associated with unusually low Taf1 occupancy (*Figure 6—figure supplement 2B*). Thus, although Taf1 and Taf7 occupancies are also moderately correlated (Pearson = 0.5; *Figure 7—figure supplement 1C*), there is a reciprocal discordance at a subset of promoters. There is a stark difference between Taf1 and Taf7 binding at non-coding RNAs: Taf7 occupancy is high at snRNAs but low at many microRNAs and long non-coding RNAs, while the Taf1 profile is the opposite (*Figure 7D* and *Figure 7—figure supplement 1D*).

Interestingly, while Taf1- and TBP-binding profiles are indistinguishable, the Taf7 peak summit in K562 cells is sometimes observed downstream of the PIC at or near the location of paused Pol II (*Figure 7E* and *Figure 7—figure supplement 2A*). In contrast, Taf1 is located only at the TSS at most genes (*Figure 7F* and *Figure 7—figure supplement 2B*). The downstream shifted peak shape is non-symmetrical, likely indicating the presence of two peaks not fully resolved: a smaller peak in the vicinity of the TSS and a larger peak downstream. In addition, as the distance between TBP and the downstream Taf7 peaks increases, the location of the paused Pol II is further downstream (*Figure 7—figure supplement 3*), providing independent evidence for a role of Taf7 in the transition to full elongation. In hESCs, the relative occupancy patterns of TBP, Taf1, and Taf7 resemble those in K562 cells (*Figure 7—figure supplement 2C,D*). The genes exhibiting downstream of Taf7 peaks are enriched for the GO categories of chromatin organization, RNA splicing, and translation, whereas the genes where Taf7 peaks are at the TSS are enriched for other classes of genes (*Figure 7—figure supplement 2E*).

## The relationship between Med26 and Taf7 varies at different gene groups

Med26 is a metazoan-specific subunit of Mediator, but it generally binds downstream of the TSS at protein-coding genes in mouse cells (*Huang et al., 2017*). Med26 interacts with both Taf7 and the P-TEFb-containing SEC complex, and at the human c-Myc and Hsp70 genes, it has been proposed to switch from binding Taf7-containing TFIID at the PIC to recruiting pTEFb to paused Pol II (*Takahashi et al., 2011*; *Lens et al., 2017*). In addition, at a subset of human snRNA genes, Med26 exchanges Taf7 for the little elongation complex, which is involved in the transcription of Pol II-dependent snRNAs (*Takahashi et al., 2011*; *Lens et al., 2017*). However, as Med26 knockdown only affects the expression of ~10 % of genes (*Takahashi et al., 2011*), it is unclear if these functions are general.

In several human and mouse cell lines, Med26 localizes downstream of the TSS and other PIC components, roughly at the position of paused Pol II (*Figure 8A*). Unlike Med1, the Med26 peak shape is bimodal, suggesting the presence of two unresolved peaks, a smaller one in the vicinity of the PIC and a larger one downstream. Interestingly, while Med26 associates with the downstream region at most genes, roughly one-third of the genes show a Med26 peak only at the PIC (*Figure 8B*). Genes showing Med26 downstream of the PIC are enriched for the GO categories of ribosome biogenesis, mRNA splicing, and translation; genes with Med26 only at the PIC are enriched for chromatin organization (*Figure 8—figure supplement 1A*). Thus, similarly to Taf7, Med26 appears to be both a component of the PIC and to act independently at a site around paused Pol II, with the relative occupancy at these locations being gene specific.

As we did not find Med26 and Taf7 datasets in the same cell line, we compared their binding profiles in different cell lines. We assumed that the behavior of GTFs would generally be similar across cell lines and that analyzing multiple human and mouse cell lines per factor would control for cell line-specific features. The genomic pattern is complex, with some genes having both downstream Med26 and Taf7 peaks, others having only downstream peaks for one of these proteins, and the remaining having no downstream peaks for either factor (*Figure 8C*). Ribosomal protein genes in both human and mouse tend to have Med26, but not Taf7, at downstream locations (*Figure 8—figure*

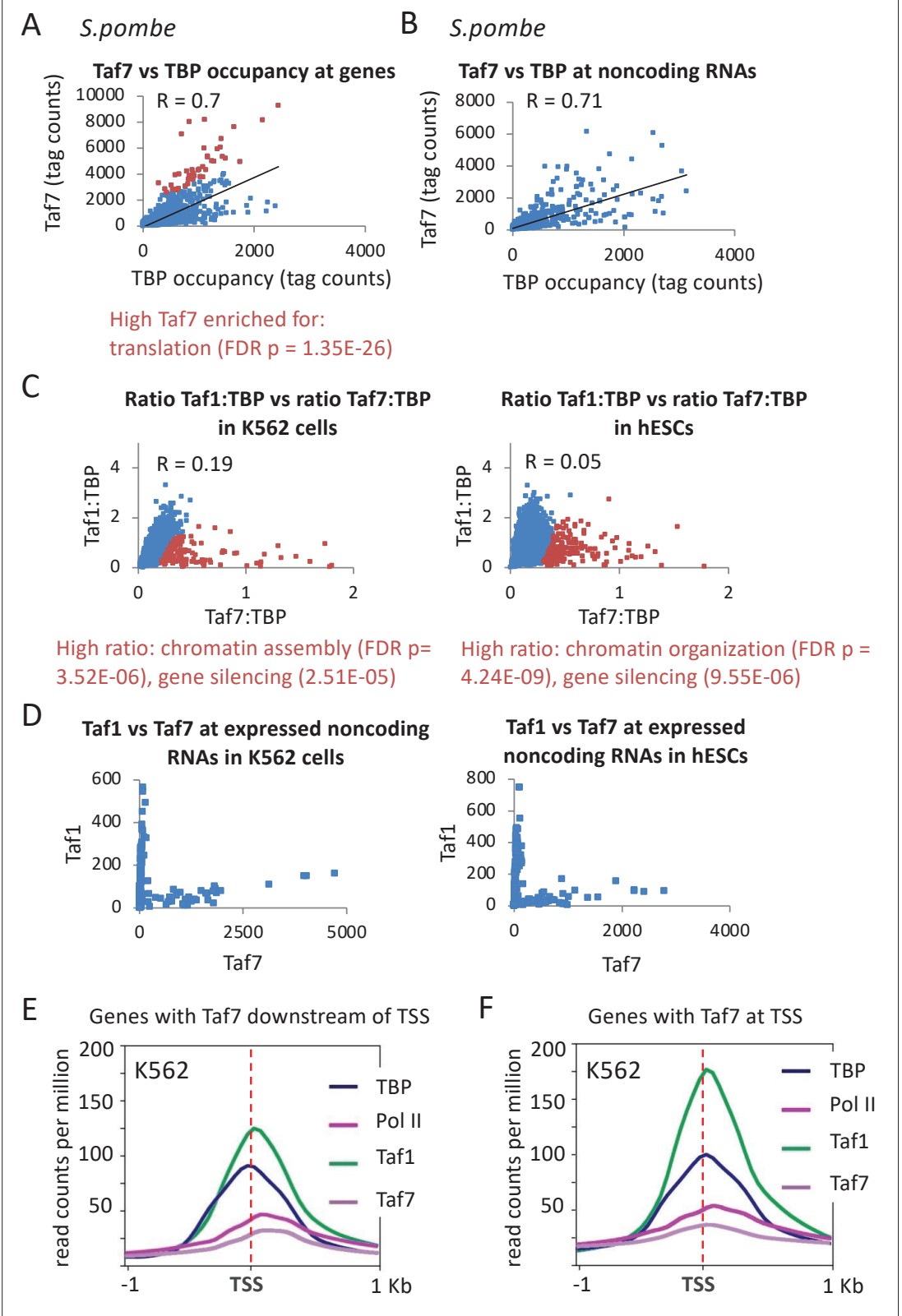

**Figure 7.** Location and relative occupancy of Taf7, Taf1, and TATA-binding protein (TBP). (**A, B**) Relative occupancy levels of TBP and Taf7 at individual mRNA and non-coding RNA promoters in *Schizosaccharomyces pombe*. Genes with relatively high Taf7:TBP ratios (red dots) are indicated along with enriched gene categories. (**C**) Relative Taf7:TBP and Taf1:TBP occupancy ratios at 10,000 most active promoters (by TBP occupancy) in the indicated human cell lines. Genes with relatively high Taf7:Taf1 ratios (red dots) are indicated along with enriched gene categories. (**D**) Relative occupancy levels

*Figure 7 continued on next page*

*Figure 7 continued*

of Taf7 and Taf1 at the 300 most expressed (by TBP occupancy) non-coding RNA promoters in the indicated cell lines. (**E**) Mean occupancies of the TBP, Taf1, Taf7, and Pol II with respect to the transcription start site (TSS) at promoters in the K562 cells with Taf7 at downstream locations. (**F**) Mean occupancies of the TBP, Taf1, Taf7, and Pol II with respect to the TSS at promoters in the K562 cells with Taf7 at the TSS. The genes in each category for **E** and **F** are presented in *Supplementary file 2*.

The online version of this article includes the following figure supplement(s) for figure 7:

**Figure supplement 1.** Relative occupancy of Taf7, Taf1, and TATA-binding protein (TBP) in human cells.

**Figure supplement 2.** Comparison of mRNA promoters in which Taf7 is located downstream or at the transcription start site (TSS) in K562 (**A, B**) and human embryonic stem cell (hESC) (**C, D**) lines.

**Figure supplement 3.** Relationship of Taf7 location with Pol II location and occupancy.

supplement 1B), whereas histone genes often have Taf7, but not Med26 at downstream locations (*Figure 8—figure supplement 1C*). Overall, Med26 and Taf7 associations are less well correlated than those of Med26 and Taf1 (*Figure 8—figure supplement 2*). At non-coding RNAs, Med26 is enriched at both the Taf1-containing and Taf7-containing groups (*Figure 8D*) discussed above, showing strong occupancy level correlation with both Taf1 and Taf7 (Pearson = 0.7 and 0.9, respectively). In sum, the relationship between Med26 and Taf7 does not appear to be generally co-dependent but rather gene-specific, suggesting potentially nuanced control of Pol II pause release.

## Discussion
### PIC stability and inactive PIC-like complexes differ at yeast and metazoan promoters

Although it is presumed that overall PIC structure and functions of individual components are conserved, there are species-specific differences in the kinetic steps in transcriptional initiation as well as the nature of stable complexes in vivo. In yeast, Mediator association with enhancers via recruitment by activator proteins is relatively stable. However, association with the promoter is transient due to near immediate phosphorylation of the Pol II CTD by TFIIH, resulting in Mediator dissociation and Pol II escape from the promoter (*Jeronimo and Robert, 2014*; *Wong et al., 2014*). Furthermore, when TFIIH-mediated phosphorylation of the Pol II CTD is inhibited, Mediator associated with the promoter lacks the kinase module. As such, the PIC in wild-type yeast cells is very short-lived, and the stable PIC-like entity is a post-escape complex that contains GTFs but lacks Mediator and Pol II (*Wong et al., 2014*). In addition, the PIC is highly unstable in yeast cells depleted of nucleotide precursors (*Petrenko et al., 2019*), unlike the situation in vitro.

In contrast, Mediator is not stably associated with activator-binding sites in three metazoan species (human, mouse, fly) at both proximal and distal locations with respect to mRNA promoters. Instead, it is stably associated with promoters in a manner that includes the kinase module. These observations are consistent with and likely explain (1) why Mediator associates with a much lower percentage of enhancers than mRNA promoters, (2) why Mediator association at enhancers rarely, if ever occurs in the absence of GTFs, and (3) why transcription from enhancers is less efficient than from mRNA promoters. Enhancers are often identified as nucleosome-depleted regions with acetylated histones (*Heintzman et al., 2007*; *Ernst et al., 2011*), which arise from activator proteins recruiting nucleosome remodeling complexes and histone acetylates. However, activator proteins at most distal enhancers do not stably recruit Mediator, even though they are able to recruit multiple chromatin-modifying activities. In contrast, yeast activators can efficiently recruit Mediator to enhancers even when PIC formation and transcription is blocked (*Knoll et al., 2018*; *Nguyen et al., 2021*). These considerations suggest that most mammalian enhancer regions lack an efficient promoter that permits a stable PIC containing Mediator.

As Mediator containing the kinase module cannot interact with Pol II or support transcription in vitro (*Elmlund et al., 2006*; *Knuesel et al., 2009*), the presence of the kinase module at metazoan promoters suggests that this is a stable, non-functional PIC-like entity. In addition, as TFIIH-mediated phosphorylation of the Pol II CTD causes dissociation of Mediator from Pol II, the high level of Mediator at metazoan promoters suggests that TFIIH-mediated phosphorylation is relatively slow and/or inefficient compared to yeast. Thus, the active PIC in metazoans, which requires Pol II to displace the

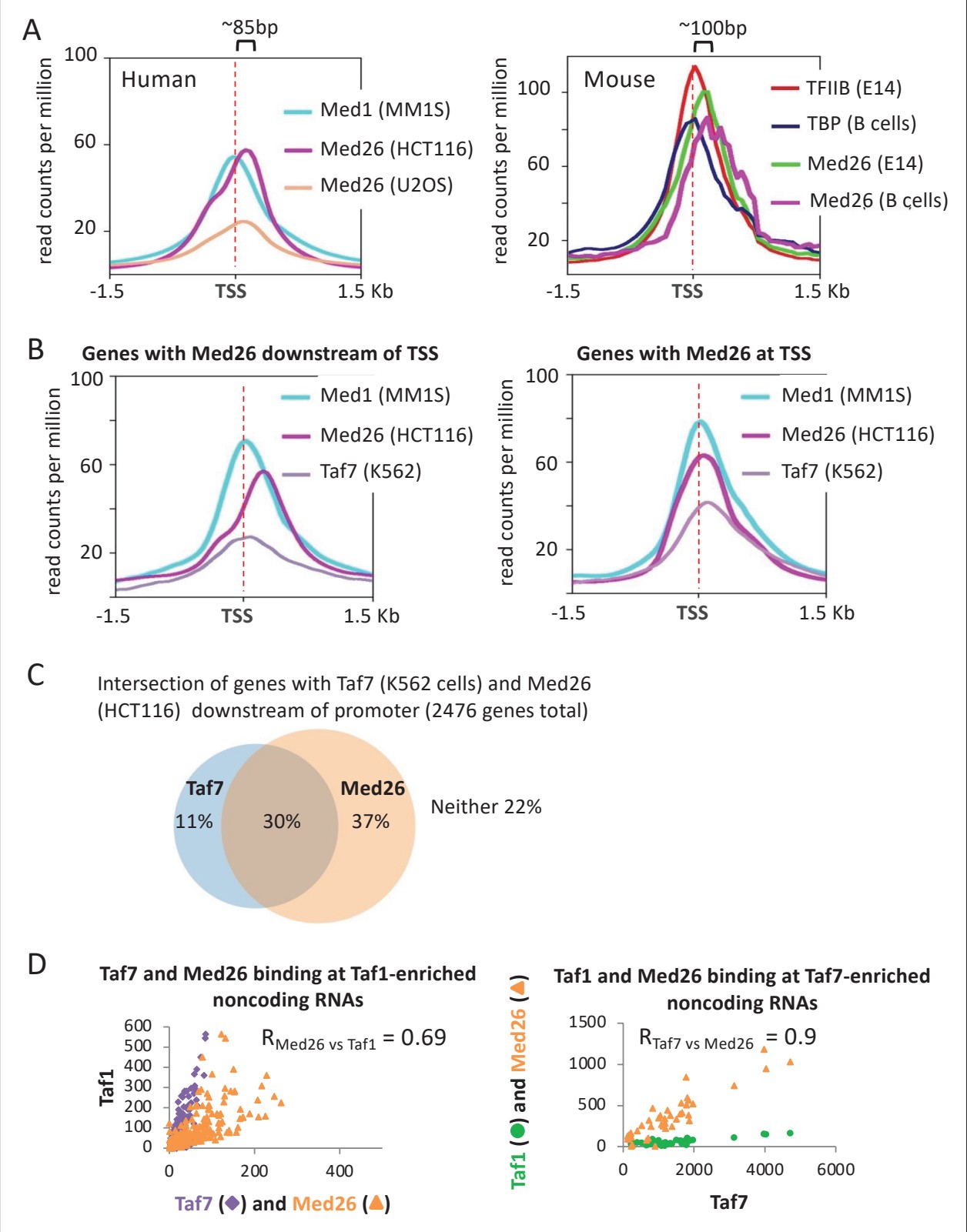

**Figure 8.** Relationship between Taf7 and Med26 occupancy in the vicinity of promoters. (**A**) Mean occupancies of the indicated proteins in the indicated mouse and human cell lines at the 10,000 most active mRNA promoters (by TATA-binding protein [TBP] occupancy). (**B**) Mean occupancies of Med1, Med26, and Taf7 in the indicated mouse and human cell lines for mRNA genes where Med26 is located downstream or at the transcription start site (TSS). (**C**) Venn diagram showing the intersection of promoters with Taf7 (K562 cells) and/or Med26 (HCT116 cells) at downstream locations. (**D**) Relative

*Figure 8 continued*

occupancies of Taf7 and Med26, versus Taf1 levels, at Taf1-enriched non-coding RNAs and relative occupancies of Taf1 and Med26 binding, versus Taf7 levels, at Taf7-enriched non-coding RNAs.

The online version of this article includes the following figure supplement(s) for figure 8:

**Figure supplement 1.** Relationship between Taf7 and Med26 occupancy in the vicinity of promoters.

**Figure supplement 2.** Relationship of Taf7 (left) or Taf1 (right) and Med26 occupancies at individual promoters in the indicated human cell lines.

Mediator kinase module from the PIC, may also be a short-lived entity, just like PICs in yeast. However, the basis for why PICs in yeast and metazoans are short-lived is different, and this is linked to the behavior of Mediator and the nature of the inactive PIC-like complexes that are stable in vivo.

The molecular components of the presumed non-functional PIC containing the Mediator kinase module are unknown. Pol II is unlikely be present in this non-functional PIC due to the lack of an interaction with the kinase-containing version of Mediator. In addition, most Pol II molecules are located at the pause site, and this likely sterically inhibits Pol II association at the PIC. Because of this steric inhibition, it is also possible that the inactive PIC-like entity might exist in two forms that differ with respect to the presence or absence of the kinase module. The absence of TFIIH in the non-functional PIC would nicely explain the presence of Mediator at the promoter, and TFIIH ChIP signals appear low when examined with multiple antibodies in different cell lines. However, it is unknown whether the low TFIIH ChIP signals reflect true occupancy or inefficient crosslinking efficiency.

## Species-specific differences in TFIID

In yeast cells, considerable evidence suggests that there are two forms of transcriptionally active TBP, namely TFIID and a TAF-independent form (*Kuras et al., 2000*; *Li et al., 2000*; *Petrenko et al., 2019*). The relative utilization of these two forms, and hence the relative occupancy of TAFs and TBP, varies among promoters. The TAF-independent form could be TBP alone, especially because free TBP is present in cell-free extracts (*Buratowski et al., 1988*). In metazoans, free TBP has never been isolated from cell-free extracts and attempts to dissociate TBP from the TFIID complex have been unsuccessful, suggesting that free TBP does not exist in appreciable quantities in vivo.

Nevertheless, as is the case in yeast, TAF:TBP occupancy ratios vary considerably among metazoan promoters suggesting that two or more transcriptionally active forms of TBP must exist. Furthermore, high and low TAF:TBP occupancy ratios are associated with different classes of genes indicating that these occupancy ratios are not due to experimental error. Instead, differences in either promoter sequence and/or activator proteins affect these gene classes. These multiple forms of transcriptionally active TBP could reflect TFIID-like complexes with different composition of TAF subunits, distinct conformations of TFIID that crosslink to the promoter with different efficiencies, or interactions with other complexes such as SAGA.

TFIID in flies differs from its yeast, human, and mouse counterparts in that it crosslinks downstream from TBP in addition to its more typical location that overlaps with TBP. Many fly promoters contain multiple promoter elements located downstream from the TATA and Initiator elements that make significant contributions to transcriptional activity (*Louder et al., 2016*; *Baumann and Gilmour, 2017*; *Vo Ngoc et al., 2019*). Such downstream promoter elements either do not exist or are less significant in yeast and mammalian cells. Presumably, the different pattern of TAF binding in flies reflects physical interactions of TAFs with the downstream promoter elements.

The behavior of Taf7 is perhaps the most dramatic species-specific difference in TFIID. In *S. pombe* (and presumably *S. cerevisiae*), Taf7 appears to behave simply as a subunit of a monolithic TFIID complex. In contrast, in at least two human cell lines, Taf7 and Taf1 occupancy is discordant at many promoters, most strikingly at those expressing non-coding RNAs.

These observations lead to several unanswered questions. First, can TFIID associate with promoters in the absence of Taf7? This seems unlikely from the structure of TFIID (*Patel et al., 2020*) unless there is another protein, such as Taf7L, that can take its place. Alternatively, low Taf7 promoters might reflect occupancy of standard TFIID, whereas high Taf7 promoters would be explained by downstream Taf7 association in addition to standard TFIID. Second, how does Taf7 associate with downstream regions near the site of the Pol II pause? It is unknown which, if any, other Tafs behave like Taf7 and whether Taf7 associates on its own or as part of a different complex. It is also unknown what aspects of paused

Pol II or some other entity is required for downstream Taf7 association, although BRD4 and P-TEFb kinase are plausible candidates. Third, what is the basis of promoter specificity of Taf7 association given that TFIID, Pol II, and P-TEFb have general roles in transcription? Differences in promoter or downstream sequences and/or gene-specific activator proteins must be involved, but the mechanism is unknown.

## A complex relationship between Med26 and Taf7

Med26 has similar properties to Taf7 in that it can also associate with downstream regions near the Pol II pause site in the apparent absence of other subunits of the major complex to which it belongs. In addition, Med26 and Taf7 physically interact, leading to suggestions that they function together to control the transition between transcriptional initiation and elongation (*Takahashi et al., 2011*; *Takahashi et al., 2015*; *Lens et al., 2017*). However, the relationship between Med26 and Taf7 is enigmatic because their associations near the Pol II pause site appear to differ considerably among genes. While this conclusion is tempered by the different cell lines used, the locations of the PIC and paused Pol II are the same in these cell lines. Furthermore, the three questions raised above for Taf7 also apply independently to Med26, thereby making the connection between these two proteins even more complicated. Whatever the mechanisms involved, they can only occur in eukaryotic species that encode Med26 and that have this unusual non-TFIID-related function of Taf7. These observations provide yet another aspect of the Pol II transcription machinery that differs considerably among eukaryotic organisms.

## Materials and methods

The list of datasets obtained from GEO (SRR numbers are the accession numbers of individual datasets) is given in *Supplementary file 1*. Sequence reads were mapped using Bowtie available through the Galaxy server (Penn State) with the following options:: *–s 0; -u 100000000; –5 11; –3 0; Phred + 33; --solexa-quals false; --int-quals false; -N 1; -L 22; -I S,1,1.15; --n-ceil L,0,0.15; --dpad 15; --gbar 4; --ignore-quals false; --no-1mm-upfront false; --local end to end; --score-min L,−0.6,−0.6; --ma 2; --mp 6,2; --np 1; --rdg 5; --rfg 5 and 3; -D 15; re-seeding 2; --seed 0; --non-deterministic false*. Normalization was performed relative to the number of mapped reads. Mean occupancy curves were generated using Galaxy deepTools (Freiburg, Germany) as well as through the Penn State Galaxy server, scaled relative to the number of mapped reads and fragment size, and expressed as counts per million mapped reads. Occupancy peaks were called using MACS available through the Penn State Galaxy server with mfold bounds at 5 and 50, bandwidth to 300–500 bp, and the FDR cutoff at 0.05. To calculate the pausing index, activity in the coding region was calculated as the mean reads 2000–4000 bases downstream of the TSS (or less for shorter genes). The determination of which genes were most active was made based on TBP levels at the promoter, or, when TBP was unavailable, the levels of another GTF.

Transcription start (TSS) and stop (TTS) coordinates for human genes were obtained through the UCSC Table Browser (https://genome.ucsc.edu/cgi-bin/hgTables), using the hg38 assembly. Coordinates for mouse were obtained from Mouse Genome Informatics (http://www.informatics.jax.org/), using the GRCm38 (mm10) assembly. For *D. melanogaster*, the Flybase (https://flybase.org/) was used to retrieve the dmel-all-r6.31 coordinates. For *S. pombe*, the Pombase (https://www.pombase.org/) was used to obtain the Schizosaccharomyces_pombe_all_chromosomes.gff3 coordinates. GO analyses were performed via the Gene Ontology Resource (http://geneontology.org/). Human enhancer coordinates for the MM1S cell line were found in *Loven et al., 2013*, which required mapping to hg18.

Boxplots were generated using Plotly Chart Studio (https://plot.ly/create/box-plot/). Venn diagrams were made using the Academo venn diagram generator (https://academo.org/demos/venn-diagram-generator/). The ChIP-seq data were visualized using the Integrated Genome Browser and with the assistance of a BED to SGR file converter, kindly provided by Zarmik Moqtaderi.

## Acknowledgements

We thank Zarmik Moqtaderi for programming scripts and discussions. This work was supported by grants to KS from the National Institutes of Health (GM30186 and GM131801).

## Additional information

### Competing interests
Kevin Struhl: Senior editor, *eLife*. The other author declares that no competing interests exist.

### Funding

| Funder | Grant reference number | Author |
| --- | --- | --- |
| National Institutes of Health | GM 30186 | Kevin Struhl |
| National Institutes of Health | GM 131801 | Kevin Struhl |

The funders had no role in study design, data collection and interpretation, or the decision to submit the work for publication.

### Author contributions
Natalia Petrenko, Conceptualization, Data curation, Formal analysis, Investigation, Methodology, Validation, Visualization, Writing - original draft, Writing - review and editing; Kevin Struhl, Conceptualization, Funding acquisition, Project administration, Supervision, Writing - original draft, Writing - review and editing

### Author ORCIDs
Kevin Struhl (iD) http://orcid.org/0000-0002-4181-7856

### Decision letter and Author response
Decision letter https://doi.org/10.7554/eLife.67964.sa1
Author response https://doi.org/10.7554/eLife.67964.sa2

## Additional files

### Supplementary files
• Supplementary file 1. Table of datasets used.
• Supplementary file 2. Table of Taf7 and Med26 downstream and at transcription start site (TSS).
• Transparent reporting form

### Data availability
All datasets and their accession numbers are listed in Supplementary file 1.

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
