## [Decision Letter]

Thank you for submitting your article "Comparison of transcriptional initiation by RNA polymerase II across eukaryotic species" for consideration by *eLife*. Your article has been reviewed by 3 peer reviewers, and the evaluation has been overseen by Naama Barkai as the Senior and Reviewing Editor. The reviewers have opted to remain anonymous.

The reviewers have discussed their reviews with one another. They are all in agreement that the paper is strong and interesting, but that some revisions are required, in particular concerning mediator association with enhancers. I attach also some notes from the communications between the reviewers which may help in preparing the revision. In particular some of the points may have several sides to them and the revision appear quite open to the discussion. Please address your point-to-point reply to all the detailed comments appended below and not to the informal discussion.

From the discussion:

Some specific comments:

1) Line 337, I suspect they mean asymmetric with respect to the TSS, but they need to clarify this. And it would definitely help to normalize the peaks so they are not so "flat"; the magnitudes for different protein ChIP signals can't be directly compared in any event.

2) I actually liked the presentation in Figures 5B-C; it is easier to discern the shape of the distribution with this than with the scatterplots. Might be helpful to add such graphs for the scatterplot data; it would clarify how outliers are defined, perhaps.

3) It would help to specify what needs fixing in Figure 6 supp1D (I agree it is not a great figure as shown). Does it need a white background or taller y-axes?

It sounds like the enhancer part should be revised substantially. That being said, this remains a quick and easy fix. They just need to use something better than CTCF, which means pretty much anything else. ESRRB in mouse ESC is a good example, but I think that they need to do more than just picking one activator. There are so many datasets available and so much information about their motifs. The data at enhancers should be plotted on the activator binding sites, if they want to make the point that Mediator is not there. Currently, they align on "…the midpoint of enhancer coordinates (Loven et al., 2013), which coincides with the location of GTFs". It is not clear what that means. Given the question they are after, it would make sense to map two ways:

1) On the activator binding sites and

2) On the TSS of the eRNA. If their conclusion holds true, Mediator should line up better on the eRNA coordinates than on the activator binding site coordinates. I also agree that in some cases (enhancers are good examples) showing heatmaps and browser views (in addition to metaplots) would be useful.

– I am not necessarily against the histogram representation (Fig5BC), although I think a scatter plot is more raw, and hence, less prone to hiding stuff. I am just concerned that they switch representation for this figure. It sounds like a uniform way of showing the data would be better.

– Regarding Fig6supp1D, I think it has several problems: Needs a y-axis that one can read (and for both panels); needs a scale bar that one can read; not clear where the genes start and end; why the grey background. The Figures, in general, are not very pleasant to the eye but it gets to another level with this one.

– I actually like Kevin's definition of promoters. It also has the merit of being clearly defined at the beginning of the paper. I am not sure what conclusion would change so drastically if using a different definition but defining the promoter as the region where the PIC assembles but not including the activator binding sites, allows discriminating between two functionally distinct elements (the promoter and the activator binding sites). Except at enhancers (where I think they are probably right but just need to show it better), the difference between yeast and metazoan is clear: in yeast, Mediator is stably associated at the UASs and resides within the PIC for a very short period; in metazoans, Mediator is not stably associated with the upstream activator binding sites (where things like SP1 or GAGA bind), but rather within the PIC (although this "PIC" most likely does not contain Pol II, based on the presence of CDK8). What bugs me a little is rather the fact that they call the UAS in yeast, "enhancers". I think that this is where the confusion may arise. UAS are not enhancers. They do not work at a distance. The UASs are more reminiscent of the promoter-proximal regulatory elements also found in higher eukaryotes (where things like SP1 or GAGA bind).

I also still think that the Med26/Taf7 part (last figure) should be removed to lean on the safe side.

The suggestion for mapping the enhancer data two ways is right on point; I think this is what should be communicated to the authors. Also his statement that yeast UASs are more like the promoter-proximal regulatory elements is on target.

With regard to the 5B-C presentation versus scatterplots, probably what should be presented to the authors is that either both formats be used or at least that scatterplots corresponding to the bar graphs also be shown, to allow cross-comparison of data.

I could go with removing the Med26-TAF7 although I found it interesting. But maybe I'm missing the point on the promoter definition. There are clearly PICs in enhancers. But Mediator is also found at some activator binding sites in enhancers without adjacent PICs. The problem is when you average, these counter examples blend in but they are very obvious on browser plots. Thus by including PICs/promoters at enhancers, the argument is not as solid. Additionally, there are typically two PICs at many mRNA encoding genes, one into the gene and one anti-sense. The anti-sense varies in strength but often is similar in abundance to PICs at enhancers. These anti-sense PICs are often in the same locations as proximal promoters so when you line these up, Mediator can align with some activators. So by using the actual gene core promoters you can easily get the results shown. But once you add enhancers and divergent promoters, the results are not as clear. The reason I picked ESRRB is because the biochemistry strongly supports a direct Mediator binding mode like many other nuclear receptors. This relationship is not as strong with other activators. Bottom line, Mediator can bind to some activators independent of PIC in mammalian cells.

The enhancer data needs more work. I think that if they align the data both ways as I suggested in my previous comments, this should do the trick. They will see better-defined aggregate peaks when aligning on the TF binding sites if your model is the prominent one. If, however, their model is correct, the peak will be better defined when aligning data on the enhancer TSSs. Showing heatmaps in addition to metaplots would help a lot, especially if you are both right and it depends on the actual enhancer (which is quite possible and would be interesting). This could then be exemplified with browser snapshots as suggested.

The fact that enhancers tend to have two relatively equal PICs should help in this analysis: aligning on the TF binding sites should lead to a valley in the center (where TFs bind) with peaks on each side (their model) or a peak in the middle (other model). Of course, the resolution may not be good enough to see the valley but at the very least, the two methods should reveal which one gives the better defines peak.

Alignment along with heat maps/browser scans should determine whether the model is correct and how generally it applies. It might help the authors if you could provide a reference or two regarding PICs sometimes being seen at enhancers and sometimes not. Or browser scans, or both.

*Reviewer #1:*

In this manuscript, Petrenko and Struhl used previously published ChIP-seq datasets for Mediator subunits and PIC components from different organisms to highlight similarities and differences in the composition of protein complexes assembled on promoters in different cells and organisms. Although purely descriptive (no perturbations are included in this study), the manuscript highlights several interesting differences between yeast (where Mediator and the PIC has been the most studied in vivo) and higher eukaryotes. The manuscript proposes ideas that, if confirmed in subsequent studies, are potentially paradigm-changing. The manuscript would be improved by strengthening the analyses on enhancers and by the removal of some aspects that are not as convincing (due to the unavailability of key datasets from the same cells).

The authors make the provocative proposal that Mediator at enhancers is not found at transcription factor binding sites (where it is classically thought to be recruited) but rather at the "promoter" of the flanking eRNAs. If true, this represents a change in paradigm. This point, however, is weak but could probably be strengthened relatively easily. This conclusion is weak for two related reasons. First, the choice of the dataset. The authors used CTCF as a representative transcription factor and compared its occupancy to that of Mediator at enhancers. It is not clear why the authors chose CTCF here. Transcription factors have been abundantly profiles by ChIP-seq in mouse and human cells, so more "canonical" TF datasets could have been used. CTCF has a TF role, indeed, but is also very well known for its role as an architectural protein at insulators and chromatin domain boundaries. These non-TF functions of CTCF may confound the analysis here. Also, the CTCF peak is not well defined. The authors claim that its occupancy does not coincide with that of Mediator. While this is clear at promoters (Figure 1-supp1C) is not obvious at enhancers (Fig1D) where CTCF has a very noisy signal (does not generate a clear peak). Hence, for both conceptual and technical reasons, CTCF appears a bad choice. Give the importance (and somewhat provocative nature) of the point they are making in this analysis, the authors should solidify their claim. Adding ChIP-seq data for other TF would most likely help a lot. Also, the authors could also map the density of TF binding motifs from databases such as Oreganno, Jasper, or TRANSFAC at enhancers.

The interplay between Taf7 and Med26 (described in Fig7) is potentially of high interest, but the fact that there exists no dataset for both these factors from the same cells, makes their comparison quite hazardous, especially when scrutinizing difference between genes (because different genes are expressed in different cells). This potentially confounds the analysis shown in Fig7. In fact, no clear conclusion came from these analyses. So, in order to refocus the manuscript on its strengths, I suggest that the section related to the relationship between Med26 and Taf7 (essentially Fig7 and its supplements) be removed.

*Reviewer #2:*

Strengths: In the past several years, application of ChIP-seq together with novel approaches for depleting essential proteins, such as those involved in transcription, has yielded new insights into the fundamental dynamics of transcriptional activation, particularly in the model organism *Saccharomyces cerevisiae*. More limited data has been obtained for metazoan organisms, and comparison of mechanisms among species is rarely reported. Here, the authors make use of publicly available datasets to ask pointed questions regarding differences and similarities in fundamental mechanisms of transcription initiation in yeast, flies, mouse and human cells. Specifically, they address Mediator dynamics, the relationship between TBP and TFIID, interactions between Tafs and promoter sequences, and interaction between Mediator and TFIID subunits at sites of paused polymerase. These analyses provide interesting insights and, just as importantly, raise new questions and point to experimentally addressable gaps in our understanding.

Weaknesses: The authors were of course constrained by the data that is actually available, which for metazoans is limited. For example, ChIP-seq data was analyzed for only two TFIID subunits in human and flies, and none for mouse. Additionally, some of the datasets used did not include replicates. These limitations make it difficult to ascertain how robust conclusions derived really are; nonetheless, most of the conclusions seem well-founded based on the data that is available.

One conclusion that could be argued is that in metazoan cells, at "so-called enhancers, Mediator is stably associated with promoters, not activator binding sites". This conclusion is based on Mediator and Cdk7, a component of TFIIH, binding at sites distinct from CTCF at enhancers in MM1S cells. However, CTCF functions mainly as insulator, not activator, so this conclusion is suspect.

Another weakness is the heavy reliance on line graphs representing average occupancies of factors, which obscures possible effects of small numbers of targets with very high occupancies. Inclusion of heat maps and more browser screenshots would be helpful. Gene ontology analyses are also presented rather superficially; such enrichments are not always as straightforwardly interpreted as simple p-values would suggest. Finally, it is not so easy to discern which datasets contributed to the individual figures.

1) In the abstract, it is stated that in human, mouse, and fly cells, Mediator with its kinase module stably associates with promoters, but not with activator-binding sites. Good evidence is presented to support this statement with regard to activator-binding sites that are close to promoter sites. However, enhancers are considered here as a second kind of promoter, rather than as primary activator-binding sites, for reasons given at the beginning of the Results section. But in comparing metazoan cells to yeast, is this the correct comparison? Given the evidence for enhancer-promoter loops in metazoans, I would think that metazoan enhancers should be considered more like UASs in yeast (i.e., the conventional view). In this case yeast and metazoan genes still behave differently, inasmuch as Mediator occupancy is observed at promoters while this is not seen in yeast. But it's not quite as stated in the manuscript.

2) There is a strong reliance on line graphs showing average occupancy of the factors examined and on scatterplots of occupancies. Line graphs can obscure cases in which a small number of data points make very strong contribution; heat maps can help in ascertaining whether this is the case. The heat maps in Figure 5—figure supplement 1 (the only ones shown) make this case, as line graphs based on these heat maps would not distinguish the different behavior of the histone cluster genes. The line graphs also appear heavily smoothed. I was unable to determine how this smoothing was done based on the terse description of methods. It appears that Bowtie, which has been superseded by Bowtie2 on Galaxy, may have been used for mapping; I could not find most of the parameters indicated in the Methods in the description of Bowtie2 on Galaxy. I'm not objecting to use of Bowtie if that was used, but a better explication of the parameters used should be provided.

It was also unclear to me how occupancy values used in the scatterplots were calculated. Were they based on simple normalized coverage (i.e. not relative to any control data)? Were they derived from MACS or calculated another way? Was anything done to ensure that aberrant data was removed? This is not always a problem but can be; as an example, TBP occupancy in yeast is very high at tRNA genes, so that Pol II transcribed genes that abut tRNA genes can give aberrantly high TBP occupancy values, depending on the details of how occupancy is calculated. Additional browser screenshots would be helpful to support conclusions in many of the figures. This is especially true when the data comes from multiple sources as it does here.

3) Gene ontology analyses should be examined more critically. Some of the FDR p-values reported are not really so impressive (> 10e-6, which for hypergeometric test is not that great). The authors should report expected (if random distribution) and observed fractions of genes in a given category found. Are the enriched categories presented in the various figures and tables the only ones found, or are there others not shown? Browser scans would be especially nice here to visualize differences, for example, between genes having high vs. low ratios of Med1/Med30 in Figure 3B.

4) It is difficult to figure out where the data used in the individual figures is from. This should be stated explicitly (i.e. references given) in the figure legends.

5) Figure 5 uses replicates of TBP ChIP as a control for distribution spread of Taf2/TBP ratios in *Drosophila*. A better control would be to compare the distribution of two different subunits of TFIID that are expected to always be present, as that could control for the use of different antibodies. Unfortunately, the data for Taf1 is from S2R cells and for Taf2 from Kc167 cells, so it is not clear that should correlate perfectly. There may be no fix for this but the limitations of the control should be recognized. The authors argue that GO enrichment is evidence for the varied Taf2/TBP ratios being meaningful, but there are artifacts that could also produce such enrichments without affecting the spread of replicate data-for example, effects due to nearby chromosomal locations of particular groups of genes, perhaps, or to particular elements preferentially neighboring genes in a GO category.

*Reviewer #3:*

This manuscript employs publicly available genomic ChIP-seq data to compare and contrast the composition of RNA Polymerase II pre-initiation complexes (PICs) among different model organisms. The data reveal interesting similarities and differences that investigators in the gene regulation field should find interesting, and that may impact the regulatory mechanisms of different classes of genes. Of interest are claimed differences in the Mediator co-activator binding in yeast versus metazoans, and an unusual mode of binding of TAF7 subunit of TFIID. Most of the conclusions were supported by the data, which employed standard analyses of ChIP-seq data with solid statistical correlation approaches. The Mediator data were limited by the absence of a more extensive analysis correlating specific activator binding in the model organisms with Mediator binding and PIC assembly, particularly at enhancers where small levels of PICs are found. This weakened the general conclusion that Mediator only binds when PICs are present in higher eukaryotic cells.

In their manuscript, Petrenko and Struhl examined how Pol II preinitiation complex (PIC) components behave differently across different eukaryotic species. By analyzing publicly available genomic data, the authors discovered how specific subunits correlate with each other and how distinct versions of Mediator/TFIID may be related to different gene categories in yeast, fly, human and mouse. For Mediator, the authors found that this co-activator complex is enriched at enhancers in yeast but mainly binds promoters in the other three species, albeit this is a debatable point (see below). The authors also argue that TFIID exists in different forms at different classes of genes. For example, the authors find Taf7 to be a special TFIID subunit in humans because it behaves differently than other TFIID subunits like TAF1. The relationship between Taf7 and the mammalian-specific protein Med26 was also studied based on the Conaway's Cell paper from a few years back.

Overall, this manuscript reports interesting discoveries that we feel will be of use to the field. For example, TFIID/Mediator forms may be gene-specific and PIC structures display differences between species. However, we are concerned with the definition of promoters used in the manuscript (see below). A different definition will change one of the main conclusions dramatically. Thus, this concern should be addressed more carefully.

1. Figure 1C. What does Med2 refer to in mammalian Mediator. Why are there Med2 (V6.5)' data in this figure? Is this Med26 or Med29? Also, we would like to see Pol II data on this graph.

2. Med1 data are shown in both Figure 1C and Figure 1 —figure supplement 1B. These data are from essentially the same cell line (E14 and E14tg2a) and plotted at the same genomic loci (10,000 most active mRNA promoters). Why is the Med1 summit located at the TSS in Figure 1C but in Figure 1 —figure supplement 1B, it is greater than 100 bp downstream of the TSS?

3. Paragraph starting on page 8, line 179. This section is confusing. First, on page 7, the authors define promoters as any genomic region bound by GTFs and Pol II (line 149-150). According to this definition, the authors appear to claim that the distal enhancers are all promoters because of Cdk7 binding as shown in Figure 1D. This led to the authors' conclusion that 'Thus, even at so-called enhancers, Mediator is stably associated with promoters' (line 186-187). However, Figure 1D shows the average data at all enhancers. There may be a portion of enhancers where there is no Cdk7 binding. It is risky to make such absolute statements. Second, the authors also conclude that Mediator is not bound at activator binding sites at enhancers (line 187). However, what is the definition of 'activator binding sites'? We did not find any data relating to this point. There are mounds of activator binding data for enhancers in many cell lines, particularly ES cells. Shouldn't the authors analyze some of these data before making such strong statements? Note that ESRRB is in an activator in mouse V6.5 ES cells. ESRRB binds Mediator directly in vitro by MS, biochemical experiments and so on. A published analysis of ESRRB at annotated enhancers shows some examples where Mediator (Med1) is bound with small amounts of PIC components including TAF1, TFIIB, and Pol II and numerous other examples where there is no evidence of a PIC. Third, what is the point of showing CTCF data? CTCF is neither a GTF nor an activator. Fourth, in the subtitle in line 158, it says 'mammalian Mediator with its kinase module … distal locations', but where are the ChIP-seq data of CKM subunits in Figure 1D for the distal locations. How can the authors make the conclusion stated in the subtitle?

4. Page 9, line 209-210. We are confused by the statement, 'Thus, unlike the case in yeast, mammalian Mediator association with promoters is not strictly correlated with association of other PIC components.' To us, this sentence sounds like that in yeast, Mediator and PIC should correlate at promoters. However, in wild type yeast, Mediator does not stably associate with promoters as shown in page 8, line 160. Please clarify.

5. Figure 5C and Figure 5—figure supplement 2A. The authors show data of TBP vs TFIIB but don't seem to describe them in the manuscript. Also, in Figure 5—figure supplement 2A, in panel with 'TBP vs TFIIB', the high and low ratio dots are not highlighted as in the other panels.

6. Page 14, line 322-323. 'However, some promoters display stronger than expected Taf7 occupancy relative to TBP and Pol II occupancy.' Where are the Pol II data shown?

7. Figure 6E and F. What is the number of plotted genes for each panel?

---

## [Author Response]

New enhancer analyses: The main concern about our enhancer analysis was valid, and we were aware of the issues raised. Analyses at enhancers are trickier than one might expect for 5 reasons. First, there are few high-quality Mediator datasets with good signals at enhancers. In retrospect and as we discuss below, this hints at the conclusion. Second, the high-quality Mediator datasets are in cell lines that typically lack datasets for PIC components. That limits the datasets we can use, and it is why we used TFIIH as the PIC component. Third, unlike promoters where the TSS is used for alignment, it is not obvious how to align enhancers. Fourth, enhancers function bidirectionally and often generate bidirectional transcripts, so there is no way to distinguish upstream from downstream. So, many analyses yield symmetric data around the alignment point, making it hard to see differences between activator binding sites and the PIC. Fifth, bidirectionality causes another problem in that the 2 divergent promoters are often too close together and can’t be resolved separately. Consequently, the GTF peaks appear to map to the center of the enhancer, which coincides with the activator peaks. In these cases, we can’t address whether Mediator is recruited by the activator or the PIC/promoter.

New Figure 2B. We circumvented the alignment and bidirectionality issues by measuring the distances between Mediator, the PIC component TFIIH, and E2F/DP1 activator summits (2 different antibodies) at both enhancer and mRNA promoters. As a control, the median distance for an individual factor in biological replicates is ~45 bp, reflecting experimental variation in peak summits. The median distances between Mediator and PIC components tested is 48 bp, confirming that Mediator coincides with the PIC. However, the median distance between Mediator and the activators tested is ~90 bp, indicating that Mediator is not associated with the activator. So, at both promoters and enhancers, Mediator localizes with the PIC, not activators bound at their target sites.

New Figure 2A. Enhancers are DNase hypersensitive with acetylated histones due to activator-mediated functions of recruiting nucleosome remodelers and histone acetylases; a typical cell line has ~150,000 such enhancers. We measured Mediator occupancy at mRNA promoters and enhancers. The top 10,000 mRNA promoters and top 10,000 enhancers have similar occupancy values. However, the median Mediator occupancy at “all” enhancers is far lower. Moreover, even this low value is mostly due to the contributions of the top 10,000 enhancers. In addition, our definition of “all” enhancers only included the ~60,000 where there was any detectable signal; hence it did not include the nearly ~100,000 enhancers where no signal was detected. This observation is striking because the vast majority of the 150,000 enhancers presumably recruit the chromatin-modifying activities (i.e. the basis for the chromatin properties of enhancers), yet recruit Mediator at very low or non-detectable levels. It is dramatically different from the situation in yeast where the activator proteins recruit normal levels of Mediator even when PIC formation and transcription is blocked.

New Figure 3B. We analyzed the relationship between Mediator, Cdk7, and Pol II at distal enhancers. The correlation of Mediator occupancy with either Cdk7 and Pol II occupancy is strong (R = 0.6) and only slightly less than the correlation between the PIC components Cdk7 and Pol II (R = 0.7) at distal enhancers and the correlations at mRNA promoters (Figure 2A). In addition, and in direct contrast to the situation in yeast, there are few, if any, enhancers in which Mediator is associated but GTFs are not.

Conclusion. These three new analyses provide independent and conclusive evidence that mammalian Mediator is stably recruited by the PIC, but not by activator proteins bound to their target sites. Of course, we believe that activator proteins interact with Mediator, but the stable association of Mediator is with the PIC in mammalian cells and with activators in yeast. Our results strongly suggest the PIC levels, Mediator occupancy, and transcription within enhancers is usually low due to the absence of a good promoter. In this view, the small subset of enhancers with high Mediator and Pol II occupancy have good promoters in the vicinity of the activator binding sites.

Reviewer #1:In this manuscript, Petrenko and Struhl used previously published ChIP-seq datasets for Mediator subunits and PIC components from different organisms to highlight similarities and differences in the composition of protein complexes assembled on promoters in different cells and organisms. Although purely descriptive (no perturbations are included in this study), the manuscript highlights several interesting differences between yeast (where Mediator and the PIC has been the most studied in vivo) and higher eukaryotes. The manuscript proposes ideas that, if confirmed in subsequent studies, are potentially paradigm-changing. The manuscript would be improved by strengthening the analyses on enhancers and by the removal of some aspects that are not as convincing (due to the unavailability of key datasets from the same cells).The authors make the provocative proposal that Mediator at enhancers is not found at transcription factor binding sites (where it is classically thought to be recruited) but rather at the "promoter" of the flanking eRNAs. If true, this represents a change in paradigm. This point, however, is weak but could probably be strengthened relatively easily. This conclusion is weak for two related reasons. First, the choice of the dataset. The authors used CTCF as a representative transcription factor and compared its occupancy to that of Mediator at enhancers. It is not clear why the authors chose CTCF here. Transcription factors have been abundantly profiles by ChIP-seq in mouse and human cells, so more "canonical" TF datasets could have been used. CTCF has a TF role, indeed, but is also very well known for its role as an architectural protein at insulators and chromatin domain boundaries. These non-TF functions of CTCF may confound the analysis here. Also, the CTCF peak is not well defined. The authors claim that its occupancy does not coincide with that of Mediator. While this is clear at promoters (Figure 1-supp1C) is not obvious at enhancers (Fig1D) where CTCF has a very noisy signal (does not generate a clear peak). Hence, for both conceptual and technical reasons, CTCF appears a bad choice. Give the importance (and somewhat provocative nature) of the point they are making in this analysis, the authors should solidify their claim. Adding ChIP-seq data for other TF would most likely help a lot. Also, the authors could also map the density of TF binding motifs from databases such as Oreganno, Jasper, or TRANSFAC at enhancers.

We agree that CTCF was a non-optimal choice and hence have removed it. Instead, we now analyze E2F and DP1, a heterodimeric activator.

The interplay between Taf7 and Med26 (described in Fig7) is potentially of high interest, but the fact that there exists no dataset for both these factors from the same cells, makes their comparison quite hazardous, especially when scrutinizing difference between genes (because different genes are expressed in different cells). This potentially confounds the analysis shown in Fig7. In fact, no clear conclusion came from these analyses. So, in order to refocus the manuscript on its strengths, I suggest that the section related to the relationship between Med26 and Taf7 (essentially Fig7 and its supplements) be removed.

The original paper recognized and discussed the problem that there are no datasets where Taf7 and Med26 are examined in the same cell line. Nevertheless, we would like to keep this analysis in the paper as it is interesting and worthy of public scrutiny. We have further softened the statement in the abstract. I note that the Taf7 and Med26 data is always compared to Pol II and TBP in the same cell line and that Pol II at most genes is in the identical location in both cell lines. Hence, although cell-type-specific effects could be involved in the observations, these are unlikely to be mediated through the PIC and paused Pol II *per se*, as these are the same in both cell lines. If the reviewers insist, we will delete this section from the paper, but we would certainly prefer that it remain.

We thank Reviewer 1 for the detailed list of suggestions in the text, almost all of which were heeded. Regarding the old Figure 6C (now 7C), we plotted TAF:TBP ratios in order to know not just the relation of one to the other but also when one of them was high relative to TBP (and thus PIC levels) and when one of them was low (i.e., depleted relative to TBP and thus PIC levels).

Reviewer #2:1) In the abstract, it is stated that in human, mouse, and fly cells, Mediator with its kinase module stably associates with promoters, but not with activator-binding sites. Good evidence is presented to support this statement with regard to activator-binding sites that are close to promoter sites. However, enhancers are considered here as a second kind of promoter, rather than as primary activator-binding sites, for reasons given at the beginning of the Results section. But in comparing metazoan cells to yeast, is this the correct comparison? Given the evidence for enhancer-promoter loops in metazoans, I would think that metazoan enhancers should be considered more like UASs in yeast (i.e., the conventional view). In this case yeast and metazoan genes still behave differently, inasmuch as Mediator occupancy is observed at promoters while this is not seen in yeast. But it's not quite as stated in the manuscript.

See enhancer section above. The comment about yeast UAS vs. enhancers addresses an important issue that I’m planning on writing about in a perspective. People often view yeast UAS and mammalian enhancers as different because yeast UAS do not work at long distances and when downstream of the promoter (Struhl, 1984; Guarente and Hoar, 1984; yes, this was a long time ago). However, an unappreciated paper (Petrascheck et al., 2015 NAR) shows that UASs can function downstream and at long distances in yeast when it is connected near the promoter via an artificial loop. Other studies indicate that the loops in mammalian cells do not occur between the enhancer and what is called the core promoter (what I call the promoter), but rather between the enhancer and activator sites near the promoter (Nolis et al., 2009 PNAS; Deng et al., 2012 Cell) and a very recent paper from Mike Carey (Sun et al., 2021 Genes Dev) indicates that loop formation is independent of the PIC. So, comparing yeast UASs and enhancers is appropriate as the difference is related to looping functions of activators, not the PIC or activation *per se*.

2) There is a strong reliance on line graphs showing average occupancy of the factors examined and on scatterplots of occupancies. Line graphs can obscure cases in which a small number of data points make very strong contribution; heat maps can help in ascertaining whether this is the case. The heat maps in Figure 5—figure supplement 1 (the only ones shown) make this case, as line graphs based on these heat maps would not distinguish the different behavior of the histone cluster genes. The line graphs also appear heavily smoothed. I was unable to determine how this smoothing was done based on the terse description of methods. It appears that Bowtie, which has been superseded by Bowtie2 on Galaxy, may have been used for mapping; I could not find most of the parameters indicated in the Methods in the description of Bowtie2 on Galaxy. I'm not objecting to use of Bowtie if that was used, but a better explication of the parameters used should be provided.It was also unclear to me how occupancy values used in the scatterplots were calculated. Were they based on simple normalized coverage (i.e. not relative to any control data)? Were they derived from MACS or calculated another way? Was anything done to ensure that aberrant data was removed? This is not always a problem but can be; as an example, TBP occupancy in yeast is very high at tRNA genes, so that Pol II transcribed genes that abut tRNA genes can give aberrantly high TBP occupancy values, depending on the details of how occupancy is calculated. Additional browser screenshots would be helpful to support conclusions in many of the figures. This is especially true when the data comes from multiple sources as it does here.

We disagree with this comment about line graphs and scatterplots, but in fact many of the desired scatterplots are already in Supplemental Figures. Line graphs are crucial for the experiments where we map the relative locations of the factors. The line graphs were done with the Galaxy Deeptools plotting tool and are not smoothed. Heat maps (and scatterplots) are inappropriate for this purpose. The Reviewer is correct that we actually used Bowtie2, and we provide more details in the Methods. As the Reviewer suspected, occupancy values were based simply on normalized coverage, of the read counts. Our new pairwise peak summit analysis (see above and new Figure 2B) is an independent way to get the same kind of information. Scatterplots are extremely useful in comparing the relative occupancies of factors at a given locus because every locus is included such that one can get an overall correlation as well as identifying “outliers” of interest, which is done throughout the paper. In fact, scatterplots would easily identify the different behavior of the histone genes. Heat maps are less useful for these purposes because the loci are ordered by some parameter, which can easily miss outliers. Of course, heat maps certainly are useful for other purposes.

Yeast tRNA genes were not considered in our analyses, and there are very few examples where the TBP signal at tRNA genes interferes with TBP at “nearby” Pol II genes (we published quite a bit on Pol III transcription in yeast and human cells). In addition, we did not identify Pol III genes at loci with low Taf:TBP occupancy ratios.

3) Gene ontology analyses should be examined more critically. Some of the FDR p-values reported are not really so impressive (> 10e-6, which for hypergeometric test is not that great). The authors should report expected (if random distribution) and observed fractions of genes in a given category found. Are the enriched categories presented in the various figures and tables the only ones found, or are there others not shown? Browser scans would be especially nice here to visualize differences, for example, between genes having high vs. low ratios of Med1/Med30 in Figure 3B.

The GO ontology analyses were corrected for multiple hypotheses, as is standard and values are defined as FDR. It is a matter of opinion about how “impressive” 10-E6 is, but clearly such values are far beyond chance. This is important because the main use of these GO analyses is to show that the “outliers” are not due to experimental variance. In addition, there are several examples in the paper of GO analyses did not yield enriched categories, and these effectively serve as controls. See also our response to comment 5.

4) It is difficult to figure out where the data used in the individual figures is from. This should be stated explicitly (i.e. references given) in the figure legends.

The figures all list the cell line and the factor, and the relevant datasets are all cited with GEO numbers.

5) Figure 5 uses replicates of TBP ChIP as a control for distribution spread of Taf2/TBP ratios in *Drosophila*. A better control would be to compare the distribution of two different subunits of TFIID that are expected to always be present, as that could control for the use of different antibodies. Unfortunately, the data for Taf1 is from S2R cells and for Taf2 from Kc167 cells, so it is not clear that should correlate perfectly. There may be no fix for this but the limitations of the control should be recognized. The authors argue that GO enrichment is evidence for the varied Taf2/TBP ratios being meaningful, but there are artifacts that could also produce such enrichments without affecting the spread of replicate data-for example, effects due to nearby chromosomal locations of particular groups of genes, perhaps, or to particular elements preferentially neighboring genes in a GO category.

The use of TBP replicates as a control for the distribution of Taf2:TBP ratios is appropriate, as these involve the same chromatin preparation and hence have identical size distributions of fragmented chromatin. Another control (unmentioned in the original paper, but now mentioned; see response to comment 5 of Reviewer 3) is the distribution of TFIIB:TBP ratios (old Figure 5C and Figure 5-supplement 2A; now Figure 6C and Figure 6-supplement 2A), which involve 2 different antibodies. Regarding the suggestion to analyze Taf1:Taf2 ratios (and aside from the lack of appropriate datasets), the use of different antibodies does not affect the relative Taf1:Taf2 ratios at promoters. Different antibodies will often have different IP efficiencies, so the absolute ratios have no meaning. However, the relative ratios among all the sites in the same samples are not affected by the different antibodies. We have used this “relative ratio” approach in many previous papers, and it is quite informative.

I think the Reviewer misunderstood our use of enriched GO categories for the varied Taf2:TBP ratios. The fact that genes in the low or high Taf2:TBP classes are enriched for certain GO categories excludes the possibility that such genes merely reflect experimental variation of ratios. As such, the results are meaningful. However, this does not (and we do not) imply any specific mechanism for why these classes exist, although I think the suggestions in this comment are unlikely. In this regard, as noted also by Reviewer 1, it is striking that some of the enriched categories in mammalian cells resemble those in yeast.

Reviewer #3:1. Figure 1C. What does Med2 refer to in mammalian Mediator. Why are there Med2 (V6.5)' data in this figure? Is this Med26 or Med29? Also, we would like to see Pol II data on this graph.

Med2 is really Med1; just a mistake. Pol II data included.

2. Med1 data are shown in both Figure 1C and Figure 1 —figure supplement 1B. These data are from essentially the same cell line (E14 and E14tg2a) and plotted at the same genomic loci (10,000 most active mRNA promoters). Why is the Med1 summit located at the TSS in Figure 1C but in Figure 1 —figure supplement 1B, it is greater than 100 bp downstream of the TSS?

We thank the reviewer for catching this discrepancy between Figure 1C and old Figure 1-supplement 1B. Indeed, the result in old Figure 1-supplement 1B is bizarre and inconsistent with every other data set indicating that Mediator is localized at the PIC. Instead, the 2 Mediator subunits tested associate roughly in the position of paused Pol II, which makes no sense. We don’t know the reason for this, but suspect it is an artifact. In this regard, my lab has encountered a few cases where a protein (totally unrelated to Mediator or the Pol II machinery) has exactly the same unexpected pattern. Furthermore, our examples and the examples in Figure 1-supplement 1B all involved ChIP-seq experiments with disuccinimidyl glutamate in addition to formaldehyde. So, we think the data in old Figure-supplement 1B is unreliable; hence this figure has been removed.

3. Paragraph starting on page 8, line 179. This section is confusing. First, on page 7, the authors define promoters as any genomic region bound by GTFs and Pol II (line 149-150). According to this definition, the authors appear to claim that the distal enhancers are all promoters because of Cdk7 binding as shown in Figure 1D. This led to the authors' conclusion that 'Thus, even at so-called enhancers, Mediator is stably associated with promoters' (line 186-187). However, Figure 1D shows the average data at all enhancers. There may be a portion of enhancers where there is no Cdk7 binding. It is risky to make such absolute statements. Second, the authors also conclude that Mediator is not bound at activator binding sites at enhancers (line 187). However, what is the definition of 'activator binding sites'? We did not find any data relating to this point. There are mounds of activator binding data for enhancers in many cell lines, particularly ES cells. Shouldn't the authors analyze some of these data before making such strong statements? Note that ESRRB is in an activator in mouse V6.5 ES cells. ESRRB binds Mediator directly in vitro by MS, biochemical experiments and so on. A published analysis of ESRRB at annotated enhancers shows some examples where Mediator (Med1) is bound with small amounts of PIC components including TAF1, TFIIB, and Pol II and numerous other examples where there is no evidence of a PIC. Third, what is the point of showing CTCF data? CTCF is neither a GTF nor an activator. Fourth, in the subtitle in line 158, it says 'mammalian Mediator with its kinase module … distal locations', but where are the ChIP-seq data of CKM subunits in Figure 1D for the distal locations. How can the authors make the conclusion stated in the subtitle?

See initial section on enhancers. Cdk7 is a subunit of TFIIH and hence is a GTF that marks the location of the PIC. Hence, by our (and Jacob and Monod’s) definition, promoters are found at so-called enhancers, this is hardly surprising given the existence of “enhancer RNAs”. Mediator and Cdk7 locations coincide, indicating that Mediator is located at the PIC in so-called enhancers. In contrast, Mediator and Cdk7 locations do not coincide with the location of activator proteins. The objection to CTCF as an activator has some merit, so we removed it from the revised manuscript and instead analyzed the E2F/DP1 activator.

4. Page 9, line 209-210. We are confused by the statement, 'Thus, unlike the case in yeast, mammalian Mediator association with promoters is not strictly correlated with association of other PIC components.' To us, this sentence sounds like that in yeast, Mediator and PIC should correlate at promoters. However, in wild type yeast, Mediator does not stably associate with promoters as shown in page 8, line 160. Please clarify.

We apologize for the confusion about the statement that Mediator and PIC components are highly correlated in yeast cells (also mentioned by Reviewer 1). This was meant to refer to conditions of Kin28 depletion/inactivation where Mediator can be seen at the promoter.

5. Figure 5C and Figure 5—figure supplement 2A. The authors show data of TBP vs TFIIB but don't seem to describe them in the manuscript. Also, in Figure 5—figure supplement 2A, in panel with 'TBP vs TFIIB', the high and low ratio dots are not highlighted as in the other panels.

We now include a sentence about the TBP:TFIIB ratio analysis, which serves as a control (see response to comment 5 of Reviewer 2). High and low dots are not highlighted in Figure 5—figure supplement 2A because this is the control experiment and there aren’t any significant high or low ratios.

6. Page 14, line 322-323. 'However, some promoters display stronger than expected Taf7 occupancy relative to TBP and Pol II occupancy.' Where are the Pol II data shown?

Correct. Pol II occupancy was not explicitly shown here so we deleted Pol II data from the text. Other analyses in the paper clearly show strong correlation of TBP and Pol II occupancy.

7. Figure 6E and F. What is the number of plotted genes for each panel?

We now include the plotted genes in old Figure 6E, F (now 7E, F) in Table 2.